# EmoSteer-TTS: Fine-Grained and Training-Free Emotion-Controllable Text-to-Speech via Activation Steering

## Abstract

Text-to-speech (TTS) has shown great progress in recent years. However, most existing TTS systems offer only coarse and rigid emotion control, typically via discrete emotion labels or a carefully crafted and detailed emotional text prompt, making fine-grained emotion manipulation either inaccessible or unstable. These models also require extensive, high-quality datasets for training. To address these limitations, we propose **EmoSteer-TTS**, a novel **training-free** approach, to achieve **fine-grained** speech emotion control (conversion, interpolation, erasure) by **activation steering**. We first empirically observe that modifying a subset of the internal activations within a flow matching-based TTS model can effectively alter the emotional tone of synthesized speech. Building on this insight, we then develop a training-free and efficient algorithm, including activation extraction, emotional token searching, and inference-time steering, which can be seamlessly integrated into a wide range of pretrained models (e.g., F5-TTS, CosyVoice2, and E2-TTS). In addition, to derive effective steering vectors, we construct a curated emotional speech dataset with diverse speakers. Extensive experiments demonstrate that EmoSteer-TTS enables fine-grained, interpretable, and continuous control over speech emotion, outperforming the state-of-the-art (SOTA). To the best of our knowledge, this is the first method that achieves training-free and continuous fine-grained emotion control in TTS. Demo samples are available at
`https://emosteer-tts-demo.pages.dev/`.

## 1 Introduction

Text-to-speech (TTS) aims to generate natural-sounding human speech from textual input (Tan et al., 2021; Xie et al., 2025). It has been widely adopted in various domains, including voice assistants, robotics, and podcast production. Emotion-controllable TTS (EC-TTS) enhances this capability by enabling control over the emotional tone of synthesized speech, making it more expressive and engaging. Fine-grained EC-TTS takes this further by allowing precise modulation of the conveyed emotion intensity in synthesized speech. Such detailed control is vital for applications requiring nuanced expressiveness, e.g., personalized storytelling (Rong et al., 2025), empathetic human-computer interaction (Wadley et al., 2022), and precise speech editing (Peng et al., 2024).

Controlling the emotional tone of synthesized speech typically requires the simultaneous manipulation of multiple characteristics, such as pitch, energy, and prosody. Independently adjusting any of these attributes often leads to undesirable artifacts. Therefore, in the literature, existing methods commonly adopt a conditional generation paradigm, including **label-based** methods that incorporate discrete emotion labels (Cho et al., 2025) and **description-based** methods that use textual emotion descriptions (Yang et al., 2025) as additional inputs to guide the speech synthesis process.

Label-based EC-TTS approaches use categorical labels (e.g., anger, happiness, fear) as an additional input to control the emotional expression during training and inference. For example, StyleTagging-TTS (Kim et al., 2021b) uses Sentence BERT (Reimers & Gurevych, 2019) to encode short phrases or keywords as emotion labels to guide the synthesis. However, such methods rely on fixed emotion labels, offering limited flexibility in control (Cong et al., 2025). Recent studies apply strength control to emotion labels. For instance, EmoSphere++ (Cho et al., 2025) converts discrete labels into the

Figure 1: Motivations of our work. (a) Existing paradigm for speech emotion control. (b) EmoSteer-TTS offers training-free, fine-grained continuous emotion control with improved interpretability.

Valence-Arousal-Dominance (VAD) vector space (Mehrabian, 1980), where the origin represents a neutral state. Both the type and intensity of emotion can be controlled by adjusting the direction and magnitude of the emotional vector. However, these methods **rely on large emotion-labeled datasets** and often **struggle to generalize** to unseen reference speech (Inoue et al., 2025).

On the other hand, description-based EC-TTS methods use textual prompts, such as "*A girl says welcome in a happy tone*", to describe the target emotion, guiding the TTS model to generate speech that aligns with the given description. For example, CosyVoice2 (Du et al., 2024) leverages textual prompts to control emotional expressiveness, enhanced via instruction fine-tuning. Similarly, EmoVoice (Yang et al., 2025) incorporates emotion descriptions into the text context to enable fine-grained emotion control. However, such methods (Guo et al., 2023; Shimizu et al., 2024; Ji et al., 2025; Li et al., 2023b) require large-scale datasets and carefully designed training procedures. Although these methods enable finer emotion manipulation, their **controllability is fundamentally limited** by the finite set of human language expressions, imposing an upper bound on control granularity. Moreover, they **exhibit instability** due to the inherent variability of textual descriptions and the stochastic nature of token sampling in the language models used for encoding.

In summary, existing methods have two limitations, i.e., **instability/poor generalization** and **coarse controllability**. The first arises from the lack of large-scale emotional speech datasets required for effective model training. The second stems from the control strategies employed in existing methods, which restrict the precision of emotion manipulation. Furthermore, the absence of exploration in emotion representations within TTS models poses challenges for researchers seeking to understand how speech emotions are encoded.

To address these limitations, we present **EmoSteer-TTS**, a training-free approach that enables fine-grained, continuous emotion control, as illustrated in Fig. 1. Specifically, we begin by analyzing the internal emotion representations of pretrained zero-shot TTS models, such as F5-TTS (Chen et al., 2025) and CosyVoice2. These models use a Diffusion Transformer (DiT) (Peebles & Xie, 2023) as the backbone and employ flow matching (Lipman et al., 2023) to generate high-fidelity mel-spectrograms. As shown in Fig. 2, we observe that only a subset of tokens, i.e., activations, within the model significantly influences the emotional tone of the synthesized speech. Building on this insight, we propose a simple yet effective algorithm to extract emotionally salient tokens, such as those associated with "sad." After identifying these tokens, we then use the difference between emotional tokens and neutral tokens to construct steering vectors for six basic emotions (Ekman, 1992). These steering vectors, combined with an adjustable strength parameter, are then used to control the synthesized emotional tone.

In summary, EmoSteer-TTS enables training-free and fine-grained emotion control, offering improved interpretability over existing approaches. The contributions of our method are:

- We present the first fine-grained and training-free EC-TTS approach by identifying and modulating internal emotion representations within existing TTS models.

- We provide new insights and enhanced interpretability for continuous EC-TTS by uncovering the emotion steering dynamics in pretrained TTS models, offering practical guidance for the design of the proposed algorithm.

- Extensive objective and subjective evaluations demonstrate the effectiveness of EmoSteer-TTS in fine-grained speech emotion control, showing its potential applicability across a wide range of pretrained TTS models.

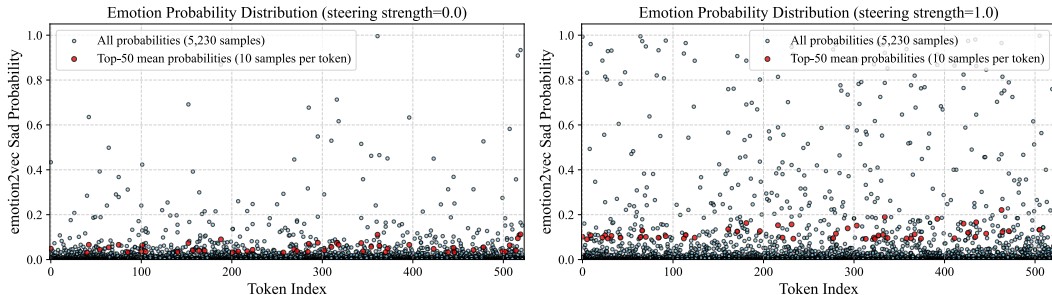

Figure 2: Adding a sadness steering vector to the activations in five DiT layers (1, 6, 11, 16, 21) of F5-TTS, conditioned on neutral speech, substantially increases the predicted sadness probability.

## 2 RELATED WORK

**Emotion-Controllable Text-to-Speech.** Unlike traditional TTS systems, e.g., VITS (Kim et al., 2021a) and VALL-E (Wang et al., 2023), that produce neutral or monotone speech, EC-TTS systems allow users to specify speech emotions, enabling more expressive and natural-sounding voices. **Label-based methods** control emotion using discrete labels (Cho et al., 2025). For instance, EmoDubber (Cong et al., 2025) uses a flow-based framework with positive/negative emotion guidance and a classifier to adjust emotion intensity. HED-TTS (Inoue et al., 2025) models hierarchical emotion distributions across speech segments, allowing multi-level intensity control. **Description-based methods** use textual prompts to specify emotions (Shimizu et al., 2024; Li et al., 2025; Ji et al., 2024; Zhou et al., 2025). PromptTTS (Guo et al., 2023) employs a BERT-based encoder to extract style from prompts and guide synthesis. VoxInstruct (Zhou et al., 2024) introduces semantic speech tokens and classifier-free guidance for fine-grained control from emotion descriptions. ControlSpeech (Ji et al., 2025) models emotional styles as Gaussian mixtures, aligning text and audio via KL divergence to enable zero-shot, controllable synthesis. Some zero-shot methods, e.g., MaskGCT (Wang et al., 2025b) and Vevo (Zhang et al., 2025), can also synthesize emotional speech, but they lack direct control and instead rely on reference speech. While these approaches have significantly advanced expressive speech synthesis, they require large-scale datasets and training.

**Activation Steering.** Activation steering aims to directly modulate the internal activations of neural networks, providing a means to exert fine-grained control over the behavior of pretrained models. Activation steering has shown great potential in the realm of LLMs. For example, it can be used to **control the behavior of LLMs**, such as enhancing the truthfulness of responses (Xiao et al., 2024; Wang et al., 2025a). Researchers can identify the mapping between the activation distributions associated with false or misleading statements and those of accurate information (Rodriguez et al., 2024). Then, during the generation process, the model's activations are steered towards the distribution representing truth, encouraging LLMs to produce more factually correct outputs (Li et al., 2023a). Activation steering can also be used to **control text-to-image (T2I) diffusion models** (Li et al., 2024; Nair et al., 2023). By modifying the activations of the diffusion model towards the distribution that corresponds to a particular style, e.g., impressionist or cubist, the model can generate images with the desired aesthetic qualities (Rodriguez et al., 2024; Brack et al., 2022). Inspired by these advances, we explore emotion representations in pretrained zero-shot TTS models and apply activation steering, offering a stable and interpretable EC-TTS method.

## 3 METHOD

### 3.1 OVERVIEW

As shown in Fig. 3, the proposed EmoSteer-TTS approach consists of three key stages. First, we compute activation differences using pairs of neutral and emotional reference speeches. Second, we identify top-$k$ emotion-relevant tokens (e.g., for "happy") to construct a steering vector and its associated weight vector. At inference time, given any unseen reference speech and text, we control the emotion of the synthesized speech by applying the steering vector with a strength parameter to modify internal activations. The proposed method is detailed in the following subsections.

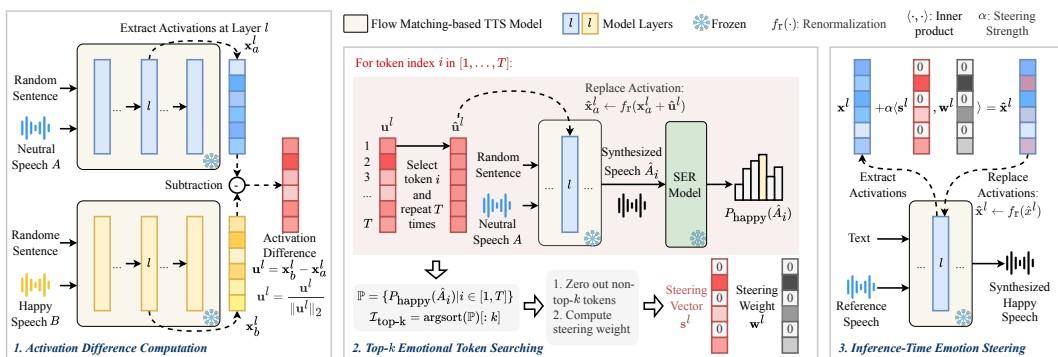

Figure 3: Overview of EmoSteer-TTS. Steering vectors and steering weights are derived from pairs of neutral and emotional reference speech. During inference, these vectors are used to modulate the activations in a TTS model, guiding it to synthesize speech that reflects the desired emotion.

## 3.2 ACTIVATION EXTRACTION

Our method focuses on zero-shot TTS models that use flow matching to synthesize mel-spectrograms. Given a pretrained TTS model with $|\mathcal{L}|$ DiT layers, we use random sentence texts along with $M$ neutral speech samples (denoted as $\mathcal{A}$) and $N$ emotional speech samples (denoted as $\mathcal{B}$) as inputs to synthesize a total of $M + N$ speech samples. For each model layer (a DiT block) $l \in \mathcal{L}$, we extract the first residual activations $\mathbf{x}_{a,i}^l$ and $\mathbf{x}_{b,j}^l$ for the synthesized speech conditioned on reference samples $A_i \in \mathcal{A}$ and $B_j \in \mathcal{B}$, respectively. The activation difference between neutral and target emotional speech at layer $l$ is defined as:

$$\mathbf{u}^l = \frac{1}{N} \sum_{j=1}^{N} \mathbf{x}_{b,j}^l - \frac{1}{M} \sum_{i=1}^{M} \mathbf{x}_{a,i}^l. \tag{1}$$

This activation difference is also known as the *difference-in-means* (Belrose et al., 2023), which can effectively extract robust feature directions. To ensure stable steering, we normalize $\mathbf{u}^l$ by dividing it by its L2 norm, resulting in a unit vector: $\mathbf{u}^l \leftarrow \frac{\mathbf{u}^l}{\|\mathbf{u}^l\|_2}$. The activation differences for all target layers to be steered are defined as $\mathcal{U} = \{\mathbf{u}^l \mid l \in \hat{\mathcal{L}}\}$, where $\hat{\mathcal{L}} \subseteq \mathcal{L}$ denotes the set of selected layers. It is worth noting that the direction of $\mathbf{u}^l$ indicates the trajectory of emotional change in the feature space, while its original magnitude reflects the extent of the transition between emotions.

Synthesized speech may vary in length. Therefore, we use nearest interpolation to align the extracted activations (token sequences) to a fixed length, which is the average activation sequence length across all $M+N$ samples. As a result, each activation has the shape $[avg\_seq\_length, hidden\_dim]$.

## 3.3 STEERING VECTOR CONSTRUCTION

After obtaining the activation difference $\mathbf{u}^l$, we select the top-$k$ tokens most relevant to the target emotion to construct the steering vector. As illustrated in Fig. 3, for each token in $\mathbf{u}^l$, we repeat token $i \in [1, 2, \ldots, T]$ $T$ times to form a new vector $\hat{\mathbf{u}}^l$. We then modify the activation $\mathbf{x}_a^l$ as follows:

$$\hat{\mathbf{x}}_a^l \leftarrow f_r(\mathbf{x}_a^l + \hat{\mathbf{u}}^l), \ f_r = \frac{\|\mathbf{x}_a^l\|_2}{\|\mathbf{x}_a^l + \hat{\mathbf{u}}^l\|_2}, \tag{2}$$

where $\mathbf{x}_a^l$ is the activation corresponding to a random sentence and a reference speech sample different from those used to compute $\mathbf{u}^l$, and $f_r$ is a function that renormalizes the modified activation to preserve the original L2 norm, which ensures more stable modification Gaintseva et al. (2025).

After the activation modification, the model synthesizes the output sample $\hat{A}_i$ corresponding to token $i$. We then use a pre-trained speech emotion recognition (SER) model, i.e., emotion2vec (Ma et al., 2024), to predict the probability that $\hat{A}_i$ corresponds to the target emotion, denoted as $P_{\text{emotion}}(\hat{A}_i)$. By computing $P_{\text{emotion}}(\hat{A}_i)$ for all tokens, we obtain the probability set:

$$\mathbb{P} = \{P_{\text{emotion}}(\hat{A}_i) | i \in [1, T]\}, \tag{3}$$

and the indices of the top-$k$ emotional tokens:

$$\mathcal{I}_{\text{top-k}} = \text{argsort}(\mathbb{P})[: k]. \tag{4}$$

Next, we zero out all non-top-$k$ tokens in $\mathbf{u}^l$ to derive the steering vector $\mathbf{s}^l$:

$$\mathbf{s}^l \leftarrow \mathbf{u}^l \odot \mathbf{m}, \ \mathbf{m}_i = \begin{cases} 1, & \text{if } i \in \mathcal{I}_{\text{top-k}} \\ 0, & \text{otherwise} \end{cases}, \tag{5}$$

where $\mathbf{m}$ is a mask vector, and $\odot$ is element-wise multiplication. To apply adaptive steering strength to each token, we compute a steering weight vector $\mathbf{w}^l$ as follows:

$$\mathbf{w}^l = \delta(\hat{\mathbb{P}}), \ \hat{\mathbb{P}} = \{P_{\text{emotion}}(\hat{A}_i) | i \in \mathcal{I}_{\text{top-k}}\}, \tag{6}$$

where $\delta$ is the Softmax function: $\delta(z_i) = \frac{e^{z_i}}{\sum_{j=1}^{k} e^{z_j}}$. Finally, we get the weighted steering vector $\hat{\mathbf{s}}^l$:

$$\hat{\mathbf{s}}^l = \langle \mathbf{s}^l, \mathbf{w}^l \rangle = \mathbf{w}_1^l \mathbf{s}_1^l + \mathbf{w}_2^l \mathbf{s}_2^l + ... + \mathbf{w}_T^l \mathbf{s}_T^l, \tag{7}$$

which can be used to steer speech emotions. Since most elements of the weighted steering vector are zero, $\hat{\mathbf{s}}^l$ lies within a subspace of the TTS model's feature space that is specifically responsible for modeling emotional tone. To ensure the efficiency of the token searching process, we simultaneously modify all selected layers at the same token indices, which can reduce the computational complexity from $\mathcal{O}(|\hat{\mathcal{L}}| \times avg\_seq\_length)$ to $\mathcal{O}(avg\_seq\_length)$.

### 3.4 FINE-GRAINED EMOTION CONTROL

In this subsection, we show how the proposed method enables fine-grained emotion control, including emotion conversion, interpolation, erasure, and composite manipulation.

**Emotion Conversion and Interpolation.** As shown in Fig. 3, given the text and reference speech, we can use the steering vector $\mathbf{s}^l$ and weight $\mathbf{w}^l$ to modify the activations in layer $l \in \hat{\mathcal{L}}$ as follows:

$$\hat{\mathbf{x}}^l = f_{\text{r}}(\mathbf{x}^l + \alpha \hat{\mathbf{s}}^l), \tag{8}$$

where $\alpha$ controls the steering strength. Note that $\hat{\mathbf{s}}^l$ has the same shape as a token, i.e., $[hidden\_dim]$. Thus, the plus sign in Eq. 8 involves an implicit broadcasting operation. Fine-grained emotion control, e.g., conversion and interpolation, can be achieved by tuning the parameter $\alpha$: when $\alpha = 0$, the emotional tone of the synthesized speech remains unchanged; when $\alpha > 0$, the emotional tone is steered toward the target emotion; and when $\alpha < 0$, it is steered in the opposite direction of the target emotion.

**Emotion Erasure.** One may wish to synthesize new speech samples using the speaking style or timbre from the reference speech while disregarding the emotional tone. Suppose the weighted steering vector $\hat{\mathbf{s}}^l$ corresponds to the emotion conveyed by the reference speech, our method achieves this by subtracting the weighted steering vector $\hat{\mathbf{s}}^l$ from the original activation $\mathbf{x}^l$, multiplied by the projection of $\hat{\mathbf{s}}^l$ onto $\mathbf{x}^l$, which can be expressed as follows:

$$\hat{\mathbf{x}}^l = f_{\text{r}}(\mathbf{x}^l - \beta(\hat{\mathbf{s}}^l \cdot \mathbf{x}^l)\hat{\mathbf{s}}^l), \tag{9}$$

where $\beta$ is the erasing strength. Eq. 9 also involves implicit broadcasting operations because $\hat{\mathbf{s}}^l$ is a single vector while $\mathbf{x}^l$ is a token sequence. Explanation of Eq. 9: Different reference speech samples may contain multiple emotions, including the target emotion at varying intensities. Our goal is to remove only the target emotion. The projection operation quantifies the intensity of the target emotion in the reference speech, while preserving all other speech characteristics.

**Composite Control.** EmoSteer-TTS also enables composite control over the emotional tone of synthesized speech. For example, given a reference speech sample, **emotion replacement** can be achieved through the following operation ($\hat{\mathbf{s}}_{\text{emo}_i}^l$ is the weighted steering vector of emotion "emo$_i$"):

$$\hat{\mathbf{x}}^l = f_{\text{r}}(\mathbf{x}^l - \beta(\hat{\mathbf{s}}_{\text{emo}_1}^l \cdot \mathbf{x}^l)\hat{\mathbf{s}}_{\text{emo}_1}^l + \alpha\hat{\mathbf{s}}_{\text{emo}_2}^l), \tag{10}$$

which replaces emotion "emo$_1$" with "emo$_2$". We can also realize **multiple emotion steering**:

$$\hat{\mathbf{x}}^l = f_{\text{r}}(\mathbf{x}^l + \alpha_1\hat{\mathbf{s}}_{\text{emo}_1}^l + \alpha_2\hat{\mathbf{s}}_{\text{emo}_2}^l + ... + \alpha_E\hat{\mathbf{s}}_{\text{emo}_E}^l), \tag{11}$$

which is particularly useful for synthesizing speech with compound emotions, such as "contempt" (disgust combined with mild anger), "pleasant surprise" (a mix of happiness and surprise), as well as more nuanced emotions like "happiness tinged with sadness" or "anger intertwined with fear".

EmoSteer-TTS enables fine-grained, continuous emotional control and supports multiple control strategies, representing the first training-free EC-TTS approach. **Appendix A** provides code snippets for the operations described above.

## 4 EXPERIMENT

### 4.1 DATASETS AND MODELS

**Datasets for Steering Vector Construction.** To obtain effective steering vectors, we construct a curated emotional speech dataset by collecting samples with clear emotional expression from multiple corpora: MSP-Podcast (Lotfian & Busso, 2017), IEMOCAP (Busso et al., 2008), RAVDESS (Livingstone & Russo, 2018), CREMA-D (Cao et al., 2014), TESS (Pichora-Fuller & Dupuis, 2020), SAVEE (Jackson & Haq, 2014), ASVP-ESD (Landry et al., 2020), CASIA (CASIA, 2023), M3ED (Zhao et al., 2022), ESD (Zhou et al., 2022), and Emo-Emilia (Zhao et al., 2025). The resulting dataset contains 6,900 utterances covering six basic emotions (anger, happiness, sadness, disgust, surprise, fear) and neutrality. Each emotion includes 1,000 samples, 500 in English and 500 in Chinese, except for fear, which has 400. The dataset includes diverse speakers with a balanced gender distribution. The construction details are provided in **Appendix B**. This dataset is used to compute activation differences between neutral and emotional speech, as defined in Eq. 1. To identify the top-$k$ tokens for each emotion, we synthesize speech using 10 random neutral ESD samples as references (5 English and 5 Chinese).

**Datasets for Inference-Time Emotion Steering.** 1) In-distribution evaluation: We sample neutral and emotional reference speeches from MSP-Podcast and ESD, which are excluded from steering vector computation. 2) Out-of-distribution (OOD) evaluation: We sample neutral speech from SeedTTS Anastassiou et al. (2024) test sets and emotional speech from EMNS Noriy et al. (2023).

**Models.** We enhance three SOTA flow matching-based TTS models (F5-TTS, CosyVoice2, E2-TTS (Eskimez et al., 2024)) using our proposed method, and compare their controllability with that of leading EC-TTS baselines, including both label-based methods with adjustable control strength (EmoSphere++, EmoDubber, HED-TTS (Inoue et al., 2025)) and description-based methods (EmoVoice, CosyVoice2, FleSpeech (Li et al., 2025)). **Appendix C** provides detailed rater information, model and hardware configurations for all experiments.

### 4.2 EMOTION CONVERSION AND INTERPOLATION

**Emotion Conversion.** We conduct emotion conversion using 100 neutral reference speech samples (50 English from MSP-Podcast and 50 Chinese from ESD), with $\alpha$=2.0 and $k$=200. We report Word Error Rate (WER), Speaker Similarity (S-SIM), Emotion Similarity (E-SIM), and Naturalness Mean Opinion Score (N-MOS, 1–5 scale, see **Appendix D** for details). WER is derived from Whisper-Large V3 (Radford et al., 2023) transcriptions. S-SIM is the cosine similarity between the embeddings of synthesized and neutral reference from a speaker embedding model (Bredin et al., 2020). E-SIM is computed as the cosine similarity between emotion2vec embeddings of synthesized speech and 100 anchor emotional samples (per emotion) from MSP-Podcast and ESD. To mitigate potential metric overfitting from emotion2vec, we also report E-SIM scores computed with SenseVoice An et al. (2024) embeddings. Since we cannot guarantee the synthesis quality of reproduced baselines, we compute their scores using demo samples for fairness. The reproduced baseline results are additionally reported in **Appendix E**. As shown in Table 1, EmoSteer-TTS achieves superior performance across multiple methods. Integrated with F5-TTS, it yields a low WER of 2.79, close to CosyVoice2 (2.53) and far better than label-based baselines. It also maintains high S-SIMs (0.66, 0.65), indicating strong speaker preservation. F5-TTS, E2-TTS, and CosyVoice2 with EmoSteer-TTS reach the top E-SIM scores, outperforming all baselines and matching FleSpeech. In N-MOS, "EmoSteer-TTS+CosyVoice2" (3.65) is close to the best (EmoVoice, 3.81), and our method consistently outperforms label-based systems. Fig. 4(a) also shows the shift in emotion probability distribution (averaged across three models) for 100 synthesized samples per emotional tone before ($\alpha$=0) and after ($\alpha$=2) emotion conversion.

Table 1: In-distribution and OOD comparison with emotion-controllable baselines.

| Method | | WER($\downarrow$) | S-SIM($\uparrow$) | E-SIM($\uparrow$) emotion2vec / SenseVoice | N-MOS($\uparrow$) | Interpolation EI-MOS($\uparrow$) | Erasure ($\beta = 2.5$) E-SIM($\uparrow$) emotion2vec / SenseVoice | EE-MOS($\uparrow$) |
|---|---|---|---|---|---|---|---|---|
| | | | | Conversion ($\alpha = 2.0$) | | | | |
| Label-based* | EmoSphere++ | 16.25 | 0.44 | 0.25 / 0.24$_{avg=0.245}$ | 3.23$_{\pm 0.81}$ | **3.50**$_{\pm 1.05}$ | - | - |
| | EmoDubber | 18.61 | 0.41 | 0.25 / 0.22$_{avg=0.235}$ | 2.47$_{\pm 1.22}$ | 2.21$_{\pm 1.08}$ | - | - |
| | HED-TTS | 13.27 | 0.52 | 0.22 / 0.26$_{avg=0.240}$ | 3.31$_{\pm 0.79}$ | 2.59$_{\pm 0.76}$ | - | - |
| Description -based* | EmoVoice | 2.91 | 0.58 | 0.27 / 0.25$_{avg=0.260}$ | **3.81**$_{\pm 0.86}$ | - | - | - |
| | CosyVoice2 | **2.53** | **0.73** | 0.24 / 0.27$_{avg=0.255}$ | **3.69**$_{\pm 1.07}$ | - | - | - |
| | FleSpeech | 9.34 | 0.54 | **0.29 / 0.26**$_{avg=0.275}$ | 3.07$_{\pm 0.75}$ | - | - | - |
| Unsteered | F5-TTS | 2.14 | 0.66 | 0.07 / 0.04$_{avg=0.055}$ | 3.79$_{\pm 0.89}$ | - | 0.03 / 0.05$_{avg=0.040}$ | 1.21$_{\pm 1.17}$ |
| | E2-TTS | 2.71 | 0.64 | 0.05 / 0.08$_{avg=0.065}$ | 3.51$_{\pm 0.94}$ | - | 0.06 / 0.02$_{avg=0.040}$ | 1.35$_{\pm 1.05}$ |
| **In-distribution evaluation on MSP-Podcast (25% en) and ESD (25% en, 50% zh)** | | | | | | | | |
| EmoSteer-TTS# (Ours) | + F5-TTS | **2.79** | 0.64 | **0.29 / 0.26**$_{avg=0.275}$ | 3.29$_{\pm 1.05}$ | **4.00**$_{\pm 0.89}$ | **0.27 / 0.25**$_{avg=0.260}$ | **4.02**$_{\pm 0.85}$ |
| | + E2-TTS | 3.28 | 0.59 | **0.28 / 0.28**$_{avg=0.280}$ | 3.31$_{\pm 0.97}$ | 3.38$_{\pm 1.09}$ | **0.24 / 0.26**$_{avg=0.250}$ | 3.63$_{\pm 1.17}$ |
| | + CosyVoice2 | 2.83 | **0.65** | 0.26 / 0.29$_{avg=0.275}$ | 3.65$_{\pm 1.08}$ | **3.56**$_{\pm 1.15}$ | **0.26 / 0.25**$_{avg=0.255}$ | 3.94$_{\pm 0.97}$ |
| **Cross-datasets (OOD) evaluation on EMNS (25% en) and SeedTT test sets (25% en, 50% zh)** | | | | | | | | |
| EmoSteer-TTS# (Ours) | + F5-TTS | **2.65** | **0.65** | 0.25 / 0.27$_{avg=0.260}$ | 3.58$_{\pm 1.04}$ | 3.46$_{\pm 1.08}$ | 0.25 / 0.22$_{avg=0.235}$ | 3.92$_{\pm 0.99}$ |
| | + E2-TTS | 3.41 | 0.55 | 0.26 / 0.25$_{avg=0.255}$ | 3.44$_{\pm 1.01}$ | **3.50**$_{\pm 0.97}$ | **0.24 / 0.27**$_{avg=0.255}$ | 3.57$_{\pm 1.03}$ |
| | + CosyVoice2 | 2.86 | **0.66** | 0.28 / 0.25$_{avg=0.265}$ | 3.49$_{\pm 1.01}$ | 3.48$_{\pm 1.27}$ | 0.23 / 0.21$_{avg=0.220}$ | **3.98**$_{\pm 0.94}$ |

*: Training-based, #: Training-free, -: Neither label-based, description-based, nor unsteered methods support interpolation or erasure. The top three results are indicated in boldface. Unsteered backbones are shown in gray for reference.

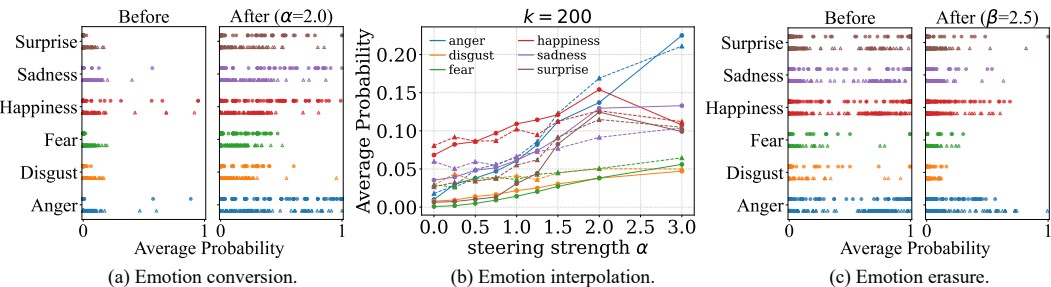

(a) Emotion conversion.    (b) Emotion interpolation.    (c) Emotion erasure.

Figure 4: Emotion steering results on MSP-Podcast and ESD. ● emotion2vec, ▲ SenseVoice.

**Emotion Interpolation.** We reuse the speech samples from the emotion conversion experiments to perform interpolation ($k$=200), gradually shifting emotional tone from neutrality to a target emotion. To assess fine-grained controllability, we report the Emotion Interpolation MOS (EI-MOS; 1–5 scale), which evaluates the alignment between target intensity and synthesized speech. Detailed criteria for EI-MOS are provided in **Appendix D**. Label-based baselines use intensity levels (e.g., 0.5 or 1.0) to control, while description-based methods, lacking intensity control, are excluded in this experiment. For fairness, baseline metrics are computed using their official demo samples. As shown in Table 1, EmoSteer-TTS achieves higher EI-MOS than label-based baselines, indicating superior capability in controlling emotional intensity. Notably, "EmoSteer-TTS+F5-TTS" obtains the highest EI-MOS of 4.00, outperforming EmoSphere++ and HED-TTS, showing better alignment with intended emotion levels. E2-TTS and CosyVoice2 variants also perform well, suggesting EmoSteer-TTS generalizes across different models. As shown in Fig. 4(b), the average predicted emotion probabilities (via emotion2vec and SenseVoice) vary smoothly with $\alpha$, illustrating EmoSteer-TTS's fine-grained controllability. However, we find that large $\alpha$ values (e.g., 3) may lead to unintelligible speech. Fig. 5(a) also illustrates smooth F0 transitions with increasing anger intensity. More examples are provided in **Appendix F**.

## 4.3 EMOTION ERASURE

We randomly select 100 unseen emotional speech samples for each type of emotion from MSP-Podcast (50 English) and ESD (50 Chinese), and erase the emotional tone using Eq. 9. We report the average E-SIM between the emotionally erased samples and 100 randomly selected neutral samples from MSP-Podcast (50 English) and ESD (50 Chinese). We also report Emotion-Erasure MOS (EE-MOS, 1-5 scale), which indicates how well the synthesized speech reflects the intended emotion erasure. Higher EE-MOS reflects better erasure performance. The standard for EE-MOS is detailed

in **Appendix D**. We set $\beta$=2.5, $k$=200 for this experiment. As shown in Table 1, our method achieves a fairly high EE-MOS score, indicating effective removal of target emotions. The decreased target emotion scores shown in Fig. 4(c) further demonstrate the emotion erasing ability. Fig. 5(b) illustrates the variation of F0 contours when gradually erasing an emotional tone. **Appendix F** provides more visualizations.

### 4.4 COMPOSITE CONTROL

**Emotion Replacement.** We use the same emotional samples from the emotion erasure experiment as reference speech for three TTS models. As defined by Eq. 10, we first remove the emotional tone of the original activation and add a target emotion. We perform six groups of replacement with $\alpha$=2, $\beta$=2.5, and $k$=200. The values in Fig. 6(a) are computed by subtracting the emotion2vec probabilities before emotion replacement from those after replacement. Each row represents a specific replacement operation (e.g., F→H denotes replacing fear with happiness), while each column indicates the predicted probability change for a given emotion. The diagonal patterns validate the success of emotion transfer, e.g., F→H shows an increase in happiness (+0.28) and a marked decrease in fear (-0.33). Similar trends are observed for other pairs, such as Su→A and H→Sa, confirming that EmoSteer-TTS effectively suppresses the original emotion and enhances the target one.

**Multi-Emotion Steering.** We use the same neutral samples from the emotion conversion experi-

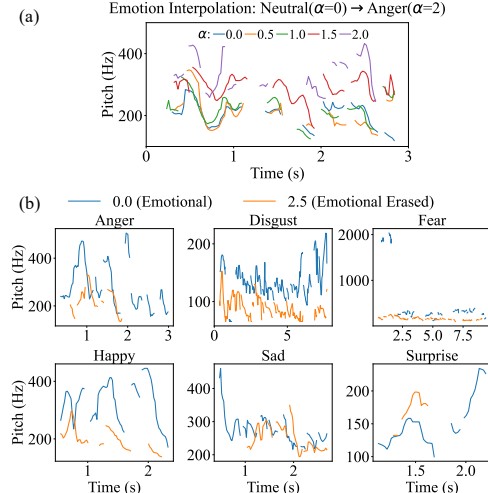

Figure 5: Visualization of F0 contours. (a) An example showing how the F0 contour varies with steering intensity; (b) The speech tone (F0 contour) becomes calmer after emotion erasure.

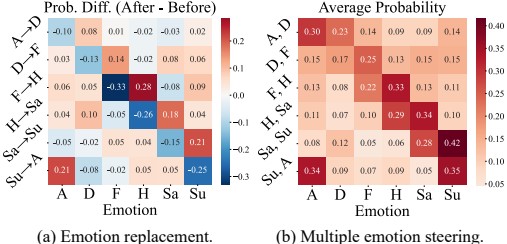

(a) Emotion replacement.    (b) Multiple emotion steering.

Figure 6: Results of composite control: (a) emotion replacement and (b) multi-emotion steering (Abbreviations: Anger, Disgust, Fear, Happiness, Sadness, Surprise)

ment as reference speech. For simplicity, this experiment simultaneously adds two emotions to the synthesized speech ($\alpha_1$=$\alpha_2$=2, $k$=200). As shown in Fig. 6(b), the predicted emotion2vec distributions align closely with the intended emotion pairs. For example, the row labeled "F, H" shows elevated probabilities for both fear (0.22) and happiness (0.33), while "Sa ,Su" leads to strong activations for sadness (0.28) and surprise (0.42). These results indicate that EmoSteer-TTS can blend multiple emotions, enabling expressive and nuanced speech synthesis beyond single-label control.

### 4.5 CROSS-DATASETS EVALUATION

Since some samples used for computing steering vectors come from the same datasets (e.g., MSP-Podcast, ESD), we also evaluate EmoSteer-TTS in an OOD setting. For emotion conversion and interpolation, we sample 100 neutral utterances from SeedTTS and 100 emotional anchors per emotion from EMNS; for emotion erasure, we use 100 emotional utterances from EMNS as references and 100 neutral anchors per emotion from SeedTTS. As shown in Table 1 (lower section), EmoSteer-TTS maintains robust performance on unseen datasets, with minimal degradation across metrics, demonstrating strong generalization beyond the steering data.

In addition to the main experiments, we report an ablation on steering corpus composition in **Appendix H.1** to investigate the influence of data quantity. We further provide correlation analyses between E-SIM and N-MOS/EE-MOS in **Appendix H.2**. Confidence intervals and significance tests for the subjective evaluations are included in **Appendix H.3** for completeness.

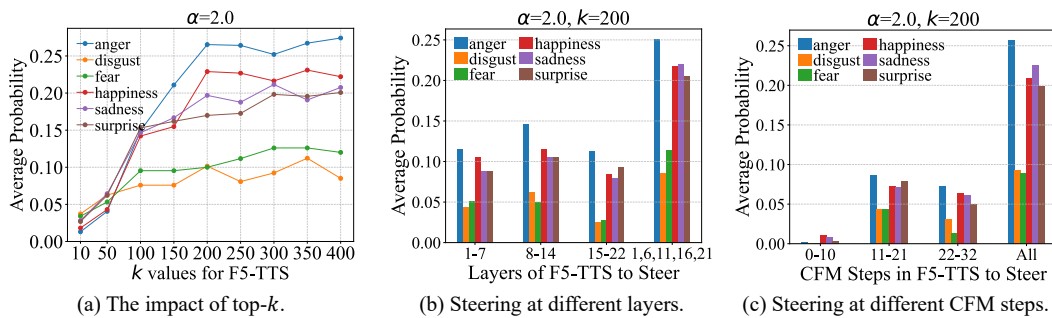

Figure 7: Analysis of emotion steering dynamics using emotion2vec predictions.

## 4.6 ANALYSIS OF EMOTION STEERING DYNAMICS

In this subsection, we analyze the emotion steering dynamics of our method. All analyses are conducted on F5-TTS, which consists of 22 DiT block layers and performs 32 flow matching steps to generate mel-spectrograms. We use the same neutral samples from the emotion conversion experiment as reference speech. We report emotion2vec emotion probabilities for all analyses.

**The Impact of Top-$k$.** The parameter $k$ determines the number of emotion-related tokens used to construct the steering vectors. A larger $k$ introduces more tokens into the steering signal, potentially capturing a broader range of emotional nuances, while a smaller $k$ focuses on the most dominant emotion features. We conduct emotion conversion with varying $k$ values (e.g., $k \in 10, 50, ..., 400$) and evaluate their impact on the synthesized emotion. Fig. 7(a) shows that increasing $k$ generally leads to higher average emotion probabilities across all categories, particularly for anger and happiness, which peak around $k$=200. Incorporating more emotion-relevant tokens enriches the steering signal, but gains plateau beyond $k = 200$ for most emotions. We therefore use $k = 200$ in all main experiments to balance expressiveness and efficiency.

**Steering Different Layers.** We examine how controlled layers affect emotion conversion by applying steering vectors $\mathbf{s}^l$ at shallow (1–7), middle (8–14), deep (15–22), and spaced layers (1, 6, 11, 16, 21). As shown in Fig. 7(b), shallow layers yield moderate emotional influence, middle layers provide slightly stronger control, and deep layers show a decline, likely focusing on acoustic details rather than emotion. Steering multiple spaced layers, however, significantly boosts probabilities across all six emotions. Overall, shallow-to-deep layers provide progressively refined control, and multi-layer steering enables the most effective emotion modulation.

**Steering Different Flow Matching Steps.** F5-TTS generates mel-spectrograms through 32 conditional flow-matching (CFM) steps. To assess the impact of steering at different stages, we apply emotion control to early (0–10), middle (11–21), late (22–32), or all steps. As shown in Fig. 7(c), early steering has little effect, while middle and late stages exert stronger influence as the spectrogram takes shape. The strongest emotion emerges when steering spans all steps, consistent with CFM's stepwise conditioning on reference speech. Therefore, we apply emotion steering across all steps in the main experiments: 32 for F5-TTS and E2-TTS, and 10 for CosyVoice2.

**Safe Steering Range.** Understanding the trade-off between steering strength $\alpha$ and audio quality is crucial for practical use. We have already reported the E-SIM variations in Fig. 4(b) for the emotion interpolation experiment. Using the same synthesized samples and newly synthesized samples with $\alpha = 2.5$, we further present the averaged in-distribution N-MOS and WER variations as a function of $\alpha$. The detailed results are shown in Tables 10, 11, and 12 in **Appendix H.4**. In summary, increasing $\alpha$ produces a highly consistent pattern across all emotions and models. For small to moderate values (up to about 1.0–1.5), N-MOS and WER remain close to the baseline. As $\alpha$ increases further, N-MOS declines and WER rises, and very large values ($\geq 2.5$) push the models outside their normal operating range, leading to distortion. This trend is nearly identical across F5-TTS, E2-TTS, and CosyVoice2, suggesting a general effect of excessive steering on model representations, likely due to shared training practices such as normalization and gradient clipping. Therefore, we recommend the following ranges for choosing $\alpha$: 1) Stable region (mild emotion): $\alpha \leq 1.0$; 2) Controlled region (stronger emotion): $1.0 < \alpha \leq 2.0$; 3) Unstable region (risk of distortion): $\alpha > 2.0$.

Table 2: Cross-lingual emotion conversion ($\alpha$=2.0, F5-TTS, token probing: emotion2vec).

| Method | WER↓ | S-SIM↑ | E-SIM↑ emotion2vec / SenseVoice | UTMOS↑ |
|---|---|---|---|---|
| English→Chinese | 92.74 | 0.21 | 0.13 / 0.08 | 2.45 |
| Chinese→English | 85.51 | 0.36 | 0.09 / 0.11 | 3.07 |

Table 3: Inference time overhead brought by EmoSteer-TTS.

| Backbone | w/o Steering (s) | Conversion (s) | Interpolation (s) | Erasure (s) |
|---|---|---|---|---|
| F5-TTS | 1.867 | 2.415 (+0.548) | 2.504 (+0.637) | 2.746 (+0.879) |
| E2-TTS | 0.942 | 1.258 (+0.316) | 1.244 (+0.302) | 1.451 (+0.509) |
| CosyVoice2 | 3.598 | 4.143 (+0.545) | 4.261 (+0.663) | 4.464 (+0.866) |

### 4.7 GENERALIZATION ANALYSIS

**The Sensitivity to SER Model for Probing.** We further evaluate the sensitivity of token probing to the choice of SER model. We replace emotion2vec with SenseVoice and report E-SIM under both embeddings to assess potential overfitting. Using the same neutral and emotional samples in the main experiments as speech prompts, we report WER, S-SIM, and E-SIM for emotion conversion and erasure. We also use UTMOS (Saeki et al., 2022) instead of N-MOS to avoid labor-intensive human evaluation. As shown in Tables 13 and 14 in **Appendix H.5**, EmoSteer-TTS shows only a very slight preference for the SenseVoice embedding space, with marginally higher E-SIM scores than under emotion2vec, indicating only mild overfitting to the SER model used for token probing. Nonetheless, human subjective results in Table 1 align with the objective metrics, confirming that EmoSteer-TTS is genuinely effective rather than overfitting a specific embedding space.

**Cross-lingual Transfer.** To assess whether a steering vector learned in one language transfers to another, we apply the precomputed English and Chinese steering vectors to the same reference samples in our in-distribution emotion conversion experiment, using the F5-TTS backbone. As shown in Table 2, cross-lingual transfer is highly limited. Applying the English vector to Chinese speech yields large WER degradation (92.74) and notably reduced S-SIM and E-SIM, indicating poor linguistic and emotional consistency. The reverse direction shows similar trends. These results suggest that emotion steering directions are largely language-specific, likely due to differences in phoneme–token mappings, prosody, and language-dependent emotional expression patterns.

We also analyze EmoSteer-TTS's robustness to noisy and reverberant prompts in **Appendix H.6**.

### 4.8 INFERENCE-TIME EFFICIENCY

To measure the computational efficiency of our method, we use the same settings as in our main experiments (conversion, interpolation, and erasure). For each type of activation steering, we employ PyTorch hooks to modify the activations during the forward pass. The additional average (per sample) inference-time overhead introduced by our method is shown in Table 3. The computational overhead is almost negligible, demonstrating the high efficiency of our methods.

## 5 CONCLUSION

We presented EmoSteer-TTS, the first training-free framework for fine-grained, continuous, and interpretable emotion control in speech synthesis. By steering a subset of internal activations in a TTS model, our method enables flexible emotional manipulation, including emotion conversion, interpolation, and erasure, without modifying or fine-tuning the pretrained TTS model. We also constructed a curated emotional speech dataset to support steering vector construction. Extensive experiments confirm that EmoSteer-TTS achieves robust, zero-shot emotion control with broad applicability, outperforming SOTA methods. The analysis also offers deeper insights into the emotion steering dynamics of flow matching-based TTS. To the best of our knowledge, this is the first fine-grained EC-TTS approach that can transform previously uncontrollable TTS models into emotionally controllable ones without any retraining, fine-tuning, and model architecture redesign.

**Limitations and Future Work.** A limitation of our method is the reliance on high-quality emotional speech samples, albeit in modest quantities, to extract effective steering vectors. In addition, strong activation steering may introduce artifacts. Future work will explore combining activation steering with learning-based approaches to mitigate these issues. We also acknowledge that whether the assumption of a linearly steerable emotion subspace holds for other architectures (e.g., VITS, VAEs, or AR models) remains an open and exciting question, which will be investigated in our future work.

## 6 Reproducibility Statement

To ensure the reproducibility of our work, we have provided comprehensive details throughout the paper and its appendices. Our proposed methodology, EmoSteer-TTS, is thoroughly described in **Section 4**, including the key algorithms for activation extraction, steering vector construction, and fine-grained control, accompanied by precise mathematical formulations. **Appendix A** further offers detailed code snippets illustrating the implementation of our core steering operations. Details regarding the datasets used for both steering vector construction and evaluation are presented in **Section 4.1**, with the specific curation and filtering process for our emotional speech dataset outlined in **Appendix B**. We also provide the code for dataset preprocessing and the processed dataset in the **Supplementary Materials**. The configurations for the TTS models (F5-TTS, E2-TTS, Cosy Voice2), including the specific layers and steps selected for steering, are detailed in **Appendix C**. The hyperparameters and experimental settings for all evaluations are specified within the relevant subsections of **Section 4**, and the criteria for our subjective evaluation metrics are defined in **Appendix D**. We will release the fully runnable code and curated dataset upon the paper's acceptance to facilitate further research.

## 7 Ethics Statement

**Possible Bias and Fairness.** Our steering vectors rely on the representations learned by SER models (emotion2vec) and the demographic distribution of our curated dataset, which may raise bias and fairness concerns. While we utilized 11 diverse corpora to ensure gender balance, the steering vectors are currently language-specific (i.e., for English and Chinese only). Future work will focus on developing language-agnostic steering vectors to ensure equitable performance across accents and dialects.

**Privacy and Data Usage.** All data used to construct the steering vectors are derived from publicly available, consented academic datasets. As a training-free method, EmoSteer-TTS does not modify model weights, eliminating the risk of accidental memorization of inference-time user data.

**Misuse and Mitigation.** We acknowledge that fine-grained emotion control increases the realism of synthesized speech, potentially raising the risk of misuse in deepfakes or social engineering. However, our method's interpretability offers a unique advantage: the steering vectors themselves act as known "signatures" of manipulation. To mitigate risks, we strongly advocate for the use of invisible audio watermarking in downstream applications. Furthermore, the "emotion erasure" capability, while potentially misuseable, also serves as a tool for removing toxic emotional cues from speech data used in training safety-aligned models.

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

## A   APPENDIX A: CODE SNIPPETS FOR FINE-GRAINED EMOTION CONTROL

For all three TTS models used in our study, i.e., F5-TTS (Chen et al., 2025), E2-TTS (Eskimez et al., 2024), and CosyVoice2 (Du et al., 2024), steering operations are implemented as hook functions. These hooks are registered either before or after the forward pass of the first residual stream in each DiT block. Code will be released upon acceptance.

### A.1   EMOTION CONVERSION AND INTERPOLATION

For emotion conversion and interpolation, we steer the activation of the first residual stream in selected DiT blocks by registering a `forward_pre_hook` that modifies the inputs before they enter the linear residual stream module. The weighted steering vector, i.e., $\hat{s}^l$ in Eq. 8, is stored in variable `steering_activations`. The steering intensity, i.e., $\alpha$ in Eq. 8, is controlled via `args.steering_strength`. The code for emotion conversion and interpolation is shown in Listing 1.

Listing 1: Emotion Conversion and Interpolation Hook

```python
def act_steering_hook(block_idx, name=None):
    """
    Create a hook function for steering activations.
    """

    def hook(module, input_args):
        if input_args and len(input_args) > 1:
            # Get the input
            (
                x,
                t,
                time,
                mask,
                rope,
                drop_audio_cond,
                drop_text,
                ref_audio_len,
            ) = input_args
            if (
                not drop_audio_cond
            ):  # If drop_audio_cond is True, no manipulation of activation values.
                step = int(time * 32) # time is a floating - point number between 0 and 1
                act = steering_activations[block_idx // 5, step, :]  # (1024)
                act = act.unsqueeze(0).repeat(
                    ref_audio_len, 1
                )  # (ref_audio_len, 1024)
                act = act.unsqueeze(0)  # (1, ref_audio_len, 1024)
                act = act.to(x.device)

                # Normalize act to unit vector
                act = act / (act.norm(p=2) + 1e-8)

                pad_len = x.size(1) - act.size(1)
                pad_tensor = torch.zeros(
                    x.size(0),
                    pad_len,
                    x.size(2),
                    dtype=x.dtype,
                    device=x.device,
                )
                act = torch.cat([act, pad_tensor], dim=1).to(x.dtype)

                # Save original norm for each sample in batch
                orig_norm = x.norm(p=2, dim=(1, 2), keepdim=True)  # (B, 1, 1)

                x = x + args.steering_strength * act

                # Rescale x to have the same norm as original x
                new_norm = x.norm(p=2, dim=(1, 2), keepdim=True) + 1e-8
                x = x * (orig_norm / new_norm)

        return (
                x,
                t,
                time,
                mask,
                rope,
                drop_audio_cond,
                drop_text,
                ref_audio_len,
            )

    return hook
```

## A.2 EMOTION ERASURE

For emotion erasure, we steer the activation of the first residual stream in selected DiT blocks by registering a `forward_pre_hook` that modifies the inputs before they enter the linear residual stream module. The weighted steering vector, i.e., $\hat{s}^l$ in Eq. 9, is stored in variable `steering_activations`. The erasing intensity, i.e., $\beta$ in Eq. 9, is controlled via `args.erasing_strength`. The code for emotion erasure is shown in Listing 2.

Listing 2: Emotion Erasure Hook

```python
def act_erasing_hook(block_idx, name=None):
```

```python
"""
Create a hook function for emotion erasure.
"""

def hook(module, input_args):
    if input_args and len(input_args) > 1:
        (
            x,  # (B, L, 1024)
            t,
            time,
            mask,
            rope,
            drop_audio_cond,
            drop_text,
            ref_audio_len,
        ) = input_args
        if (
            not drop_audio_cond
        ):
            step = int(time * 32)
            act = steering_activations[block_idx // 5, step, :]  # (1024)
            act = act.to(x.dtype).to(x.device)

            # Normalize act to unit vector
            act = act / (act.norm(p=2) + 1e-8)

            projection = torch.matmul(
                act.unsqueeze(0),  # (1, 1024)
                x[:, :ref_audio_len, :].transpose(
                    1, 2
                ),  # (B, ref_audio_len, 1024)
            ).transpose(
                1, 2
            )  # (B, ref_audio_len, 1)

            pad_len = x.size(1) - ref_audio_len
            padded_projection = torch.cat(
                [
                    projection,
                    torch.zeros(
                        x.size(0),
                        pad_len,
                        1,
                        dtype=x.dtype,
                        device=x.device,
                    ),
                ],
                dim=1,
            )

            act = act.unsqueeze(0).repeat(
                ref_audio_len, 1
            )  # (ref_audio_len, 1024)
            act = act.unsqueeze(0)  # (1, ref_audio_len, 1024)

            pad_tensor = torch.zeros(
                x.size(0),
                pad_len,
                x.size(2),
                dtype=x.dtype,
                device=x.device,
            )
            act = torch.cat([act, pad_tensor], dim=1)

            # Save original norm for each sample in batch
            orig_norm = x.norm(p=2, dim=(1, 2), keepdim=True)  # (B, 1, 1)

            x = x - args.erasing_strength * padded_projection * act

            # Rescale x to have the same norm as original x
            new_norm = x.norm(p=2, dim=(1, 2), keepdim=True) + 1e-8
            x = x * (orig_norm / new_norm)

        return (
            x,
            t,
            time,
            mask,
            rope,
            drop_audio_cond,
            drop_text,
```

```
83                        ref_audio_len,
84                )
85
86        return hook
```

## A.3 EMOTION REPLACEMENT

For emotion replacement, we steer the activation of the first residual stream in selected DiT blocks by registering a `forward_pre_hook`, which modifies the inputs before they enter the linear residual stream module. The weighted steering vectors for emotion 1 and emotion 2, i.e., $\hat{s}^l_{emo1}$ and $\hat{s}^l_{emo2}$ in Eq. 10, are stored in the variable `steering_activations_1` and variable `steering_activations_2`, respectively. The erasing and steering intensities, i.e., $\beta$ and $\alpha$ in Eq. 10, are controlled by variables `args.erasing_strength` and `args.steering_strength`, respectively. The implementation of emotion replacement is provided in Listing 3.

Listing 3: Emotion Replacement Hook

```
1    def act_replacement_hook(block_idx, name=None):
2        """
3        Create a hook function for emotion replacement.
4        """
5
6        def hook(module, input_args):
7            if input_args and len(input_args) > 1:
8                (
9                    x,   # (B, L, 1024)
10                   t,
11                   time,
12                   mask,
13                   rope,
14                   drop_audio_cond,
15                   drop_text,
16                   ref_audio_len,
17               ) = input_args
18               if (
19                   not drop_audio_cond
20               ):
21                   step = int(time * 32)
22                   act1 = steering_activations_1[block_idx // 5, step, :]  # (1024)
23                   act1 = act.to(x.dtype).to(x.device)
24
25                   # Normalize act to unit vector
26                   act1 = act1 / (act1.norm(p=2) + 1e-8)
27
28                   projection = torch.matmul(
29                       act1.unsqueeze(0),  # (1, 1024)
30                       x[:, :ref_audio_len, :].transpose(
31                           1, 2
32                       ),  # (B, ref_audio_len, 1024)
33                   ).transpose(
34                       1, 2
35                   )  # (B, ref_audio_len, 1)
36
37                   pad_len = x.size(1) - ref_audio_len
38                   padded_projection = torch.cat(
39                       [
40                           projection,
41                           torch.zeros(
42                               x.size(0),
43                               pad_len,
44                               1,
45                               dtype=x.dtype,
46                               device=x.device,
47                           ),
48                       ],
49                       dim=1,
50                   )
51
52                   act1 = act1.unsqueeze(0).repeat(
53                       ref_audio_len, 1
54                   )  # (ref_audio_len, 1024)
55                   act1 = act1.unsqueeze(0)  # (1, ref_audio_len, 1024)
56
57                   pad_tensor = torch.zeros(
58                       x.size(0),
59                       pad_len,
```

```
60                          x.size(2),
61                          dtype=x.dtype,
62                          device=x.device,
63                      )
64                      act1 = torch.cat([act1, pad_tensor], dim=1)
65
66                      act2 = steering_activations_2[block_idx // 5, step, :]  # (1024)
67                      act2 = act2.unsqueeze(0).repeat(
68                          ref_audio_len, 1
69                      )  # (ref_audio_len, 1024)
70                      act2 = act2.unsqueeze(0)  # (1, ref_audio_len, 1024)
71                      act2 = act2.to(x.device)
72
73                      # Normalize act2 to unit vector
74                      act2 = act2 / (act2.norm(p=2) + 1e-8)
75
76                      pad_len = x.size(1) - act2.size(1)
77                      pad_tensor = torch.zeros(
78                          x.size(0),
79                          pad_len,
80                          x.size(2),
81                          dtype=x.dtype,
82                          device=x.device,
83                      )
84                      act2 = torch.cat([act2, pad_tensor], dim=1).to(x.dtype)
85
86                      # Save original norm for each sample in batch
87                      orig_norm = x.norm(p=2, dim=(1, 2), keepdim=True)  # (B, 1, 1)
88
89                      x = x - args.erasing_strength * padded_projection * act1 + args.
                              steering_strength * act2
90
91                      # Rescale x to have the same norm as original x
92                      new_norm = x.norm(p=2, dim=(1, 2), keepdim=True) + 1e-8
93                      x = x * (orig_norm / new_norm)
94
95                  return (
96                      x,
97                      t,
98                      time,
99                      mask,
100                     rope,
101                     drop_audio_cond,
102                     drop_text,
103                     ref_audio_len,
104                 )
105
106      return hook
```

## A.4 MULTIPLE EMOTION STEERING

For multiple emotion steering, we steer the activation of the first residual stream in selected DiT blocks by registering a `forward_pre_hook` that modifies the inputs before they enter the linear residual stream module. The weighted steering vectors for emotions 1 and 2, i.e., $\hat{s}^l_{emo1}$ and $\hat{s}^l_{emo2}$ in Eq. 11, are stored in variable `steering_activations_1` and variable `steering_activations_2`, respectively. The steering intensities for the two emotions, i.e., $\alpha_1$ and $\alpha_2$ in Eq. 11, are controlled via `args.steering_strength_1` and `steering_strength_2`, respectively. The code for multiple emotion steering is shown in Listing 4.

Listing 4: Multiple Emotion Steering Hook

```
1  def act_multi_steering_hook(block_idx, name=None):
2      """
3      Create a hook function for multiple emotion steering.
4      """
5
6      def hook(module, input_args):
7          if input_args and len(input_args) > 1:
8              # Get the input
9              (
10                 x,
11                 t,
12                 time,
13                 mask,
14                 rope,
15                 drop_audio_cond,
```

```
16                      drop_text,
17                      ref_audio_len,
18              ) = input_args
19              if (
20                      not drop_audio_cond
21              ):
22                      step = int(time * 32)
23                      act1 = steering_activations_1[block_idx // 5, step, :]   # (1024)
24                      act1 = act1.unsqueeze(0).repeat(
25                          ref_audio_len, 1
26                      )  # (ref_audio_len, 1024)
27                      act1 = act1.unsqueeze(0)  # (1, ref_audio_len, 1024)
28                      act1 = act1.to(x.device)
29
30                      # Normalize act to unit vector
31                      act1 = act1 / (act1.norm(p=2) + 1e-8)
32
33                      pad_len = x.size(1) - act1.size(1)
34                      pad_tensor = torch.zeros(
35                          x.size(0),
36                          pad_len,
37                          x.size(2),
38                          dtype=x.dtype,
39                          device=x.device,
40                      )
41                      act1 = torch.cat([act1, pad_tensor], dim=1).to(x.dtype)
42
43                      act2 = steering_activations_2[block_idx // 5, step, :]   # (1024)
44                      act2 = act2.unsqueeze(0).repeat(
45                          ref_audio_len, 1
46                      )  # (ref_audio_len, 1024)
47                      act2 = act2.unsqueeze(0)  # (1, ref_audio_len, 1024)
48                      act2 = act2.to(x.device)
49
50                      # Normalize act to unit vector
51                      act2 = act2 / (act2.norm(p=2) + 1e-8)
52
53                      pad_len = x.size(1) - act2.size(1)
54                      pad_tensor = torch.zeros(
55                          x.size(0),
56                          pad_len,
57                          x.size(2),
58                          dtype=x.dtype,
59                          device=x.device,
60                      )
61                      act2 = torch.cat([act2, pad_tensor], dim=1).to(x.dtype)
62
63                      # Save original norm for each sample in batch
64                      orig_norm = x.norm(p=2, dim=(1, 2), keepdim=True)  # (B, 1, 1)
65
66                      x = x + args.steering_strength_1 * act1 + args.steering_strength_2 * act2
67
68                      # Rescale x to have the same norm as original x
69                      new_norm = x.norm(p=2, dim=(1, 2), keepdim=True) + 1e-8
70                      x = x * (orig_norm / new_norm)
71
72              return (
73                      x,
74                      t,
75                      time,
76                      mask,
77                      rope,
78                      drop_audio_cond,
79                      drop_text,
80                      ref_audio_len,
81              )
82
83      return hook
```

## B   APPENDIX B: DATASET CONSTRUCTION

To ensure the effectiveness of the steering vectors, we curate an emotional speech dataset by collecting and filtering audio samples with clearly distinguishable emotional tones from multiple existing corpora, including MSP-Podcast (Lotfian & Busso, 2017), IEMOCAP (Busso et al., 2008), RAVDESS (Livingstone & Russo, 2018), CREMA-D (Cao et al., 2014), TESS (Pichora-Fuller & Dupuis, 2020), SAVEE (Jackson & Haq, 2014), ASVP-ESD (Landry et al., 2020), CASIA (CASIA,

Table 4: The details of model configuration for activation steering.

| Model | # Layers | # CFM Steps | Steered Layers | Steered Activations in Each Layer |
|---|---|---|---|---|
| F5-TTS | 22 | 32 | Every 5 layers starting from layer 1 | The first residual stream |
| E2-TTS | 8 | 32 | Every 3 layers starting from layer 1 | The first residual stream |
| CosyVoice2 | 56 | 10 | Every 5 layers starting from layer 1 | The first residual stream |

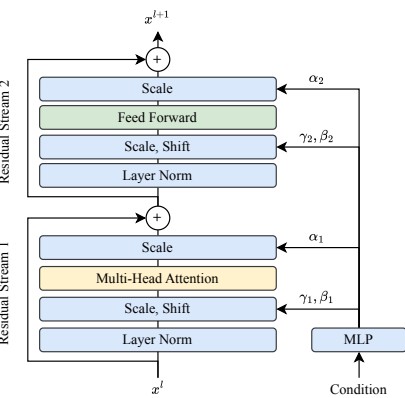

Figure 8: A DiT block in a CFM-based TTS model.

2023), M3ED (Zhao et al., 2022), ESD (Zhou et al., 2022), and Emo-Emilia (Zhao et al., 2025). The quality filtering process involves the following steps:

1. We use `librosa` (McFee et al., 2015) to remove utterances that are either too short ($<2$s) or too long ($>20$s).

2. We further filter out samples exhibiting excessive silence ($>30\%$) or a low signal-to-noise ratio (SNR) ($<10$dB), also using `librosa`.

3. We use a SER model, emotion2Vec, to eliminate samples with low recognition confidence ($<0.6$), retaining only those predicted as ground truth labels.

4. Finally, we perform a manual inspection on 50% of the data to ensure overall dataset quality.

The resulting dataset covers a broad range of speakers, emotions, and speaking styles, providing a robust foundation for learning and evaluating fine-grained emotion steering in text-to-speech synthesis. Data and code will be released upon acceptance.

## C APPENDIX C: CONFIGURATIONS

### C.1 MODEL CONFIGURATIONS

We steer three pretrained conditional flow matching (CFM)-based TTS models, i.e., F5-TTS, E2-TTS, and CosyVoice2, in our main experiments. As illustrated in Fig.8, we apply steering to the first residual stream at each layer of these models. The detailed configurations of the models are provided in Table 4. emotion2vec and SenseVoice checkpoints are downloaded from their official repos[1][2].

### C.2 HARDWARE AND SOFTWARE CONFIGURATIONS

All experiments were conducted on a server equipped with 8× NVIDIA RTX 6000 Ada GPUs (48GB each) and 2× Intel(R) Xeon(R) Platinum 8375C CPUs (2.9GHz, 32 cores each), with a total of

---

[1]https://huggingface.co/emotion2vec/emotion2vec_plus_large
[2]https://huggingface.co/FunAudioLLM/SenseVoiceSmall

256GB of RAM. The operating system is Ubuntu 20.04.6 LTS. All code was executed in Conda environments. The relevant software libraries and frameworks for each model (F5-TTS, E2-TTS, CosyVoice2) are described in their GitHub repositories[3][4][5].

### C.3 RATERS' INFORMATION AND INTER-RATER RELIABILITY

30 raters participated in the human evaluation for our main experiments. All raters were either master's or PhD students. We adopt Percent Agreement (Gwet, 2014) as a more appropriate measure of reliability for the human evaluation of synthesized samples. The results show a Top-2 Box Agreement of 88.1%, meaning that the vast majority of ratings fell within the 4 (Good) or 5 (Excellent) categories. Furthermore, the raters demonstrated high consistency in their qualitative judgment, with negligible divergence on the acceptable range.

## D APPENDIX D: OBJECTIVE EVALUATION METRICS

### D.1 NATURALNESS MEAN OPINION SCORE

The Naturalness Mean Opinion Score (N-MOS) evaluates the perceived naturalness of synthesized speech on a 5-point Likert scale. Participants are asked to rate each utterance based solely on how natural and human-like it sounds, regardless of its emotional expressiveness or content accuracy. The scale is defined as follows:

- 5 — Completely natural: indistinguishable from real human speech.
- 4 — Mostly natural: minor artifacts but still sounds largely human.
- 3 — Moderately natural: noticeable synthetic artifacts, but intelligible.
- 2 — Barely natural: speech is intelligible but sounds clearly robotic.
- 1 — Not natural at all: heavily distorted or unnatural-sounding.

Each utterance is evaluated by multiple annotators, and the final N-MOS is computed as the average score across all evaluations.

### D.2 EMOTION INTERPOLATION MEAN OPINION SCORE

The Emotion Interpolation Mean Opinion Score (EI-MOS) assesses the system's ability to smoothly interpolate between two emotional styles. For each interpolation sequence (e.g., neutral $\rightarrow$ angry), raters listen to a series of utterances generated with gradually increasing emotion intensity and judge how naturally and smoothly the emotional change is conveyed. Raters are instructed to focus on the continuity and consistency of emotional expression rather than the naturalness or correctness of individual utterances. The scoring scale is as follows:

- 5 — Emotion transition is smooth and realistic throughout the sequence.
- 4 — Emotion changes are mostly smooth, with minor inconsistencies.
- 3 — Some transitions feel abrupt or inconsistent.
- 2 — Transitions are disjointed, or emotion interpolation feels unnatural.
- 1 — No meaningful emotion interpolation perceived.

Each interpolation sequence is rated by multiple annotators, and the EI-MOS is reported as the average of all scores.

---

[3] https://github.com/SWivid/F5-TTS
[4] https://github.com/lucidrains/e2-tts-pytorch
[5] https://github.com/FunAudioLLM/CosyVoice

### D.3 EMOTION ERASURE MEAN OPINION SCORE

The Emotion Erasure Mean Opinion Score (EE-MOS) evaluates the effectiveness of emotion removal from synthesized speech. Specifically, it measures how well the target emotion has been erased, with the desired outcome being emotionally neutral and natural-sounding speech. Annotators are instructed to assess whether the emotional content of the original utterance has been successfully suppressed or removed. The rating is based on a 5-point scale:

- 5 — *Emotion fully removed*: The target emotion is completely erased; the speech sounds emotionally neutral and natural, with no detectable emotional cues.

- 4 — *Emotion mostly removed*: Only faint traces of the original emotion remain; the speech is close to neutral.

- 3 — *Emotion partially removed*: The emotional intensity is reduced, but the target emotion is still clearly noticeable.

- 2 — *Emotion barely removed*: The emotional expression remains strong; only minimal reduction is observed.

- 1 — *Emotion not removed*: The original emotional tone persists fully or is even unintentionally enhanced.

Each utterance is evaluated independently by multiple listeners, and the EE-MOS is calculated as the average of all individual scores. A higher EE-MOS indicates a more effective erasure of the target emotion.

Table 5: Unguaranteed reproduced results of open-source baselines.

| Method | | Conversion ($\alpha = 2.0$) | | | | Interpolation | Erasure ($\beta = 2.5$) | |
|---|---|---|---|---|---|---|---|---|
| | | WER($\downarrow$) | S-SIM($\uparrow$) | E-SIM($\uparrow$) emotion2vec / SenseVoice | N-MOS($\uparrow$) | EI-MOS($\uparrow$) | E-SIM($\uparrow$) emotion2vec / SenseVoice | EE-MOS($\uparrow$) |
| **In-distribution evaluation on MSP-Podcast (25% en) and ESD (25% en, 50% zh)** | | | | | | | | |
| Label-based* | EmoSphere++ | 37.29 | 0.21 | 0.14 / 0.11$_{avg=0.125}$ | 2.14$_{\pm0.91}$ | 2.41$_{\pm0.83}$ | - | - |
| | EmoDubber | 65.93 | 0.16 | 0.08 / 0.05$_{avg=0.065}$ | 1.07$_{\pm0.94}$ | 1.13$_{\pm1.02}$ | - | - |
| Description -based* | EmoVoice | 5.31 | 0.48 | 0.22 / 0.19$_{avg=0.205}$ | 3.26$_{\pm1.22}$ | - | - | - |
| | CosyVoice2 | 2.71 | 0.69 | 0.23 / 0.25$_{avg=0.240}$ | **3.66**$_{\pm1.17}$ | - | - | - |
| Unsteered | F5-TTS | 2.14 | 0.66 | 0.07 / 0.04$_{avg=0.055}$ | 3.79$_{\pm0.89}$ | - | 0.03 / 0.05$_{avg=0.040}$ | 1.21$_{\pm1.17}$ |
| | E2-TTS | 2.71 | 0.64 | 0.05 / 0.08$_{avg=0.065}$ | 3.51$_{\pm0.94}$ | - | 0.06 / 0.02$_{avg=0.040}$ | 1.35$_{\pm1.05}$ |
| EmoSteer-TTS# (Ours) | + F5-TTS | **2.79** | 0.64 | **0.29 / 0.26**$_{avg=0.275}$ | 3.29$_{\pm1.05}$ | **4.00**$_{\pm0.89}$ | **0.27 / 0.25**$_{avg=0.260}$ | **4.02**$_{\pm0.85}$ |
| | + E2-TTS | 3.28 | 0.59 | **0.28 / 0.28**$_{avg=0.280}$ | 3.31$_{\pm0.97}$ | 3.38$_{\pm1.09}$ | **0.24 / 0.26**$_{avg=0.250}$ | 3.63$_{\pm1.17}$ |
| | + CosyVoice2 | 2.83 | **0.65** | **0.26 / 0.29**$_{avg=0.275}$ | 3.65$_{\pm1.08}$ | 3.56$_{\pm1.15}$ | **0.26 / 0.25**$_{avg=0.255}$ | 3.94$_{\pm0.97}$ |
| **Cross-datasets (OOD) evaluation on EMNS (25% en) and SeedTT test sets (25% en, 50% zh)** | | | | | | | | |
| EmoSteer-TTS# (Ours) | + F5-TTS | **2.65** | 0.65 | 0.25 / 0.27$_{avg=0.260}$ | 3.58$_{\pm1.04}$ | 3.46$_{\pm1.08}$ | 0.25 / 0.22$_{avg=0.235}$ | 3.92$_{\pm0.99}$ |
| | + E2-TTS | 3.41 | 0.55 | 0.26 / 0.25$_{avg=0.255}$ | 3.44$_{\pm1.07}$ | **3.50**$_{\pm0.97}$ | **0.24 / 0.27**$_{avg=0.255}$ | 3.57$_{\pm1.03}$ |
| | + CosyVoice2 | 2.86 | **0.66** | 0.28 / 0.25$_{avg=0.265}$ | 3.49$_{\pm1.01}$ | 3.48$_{\pm1.27}$ | 0.23 / 0.21$_{avg=0.220}$ | 3.98$_{\pm0.94}$ |

*: Training-based, #: Training-free, -: Unsupported operation.
The top three results are indicated in boldface. Unsteered backbones are shown in gray for reference.

## E APPENDIX E: REPRODUCED BASELINE RESULTS

This section recomputes baselines under a controlled protocol, using the same text prompts, reference speeches, and evaluation scripts as in the in-distribution evaluation on MSP-Podcast and ESD. We report results only for open-source methods, as the reproduced quality cannot be guaranteed. Therefore, the results in Table 5 are provided for reference only.

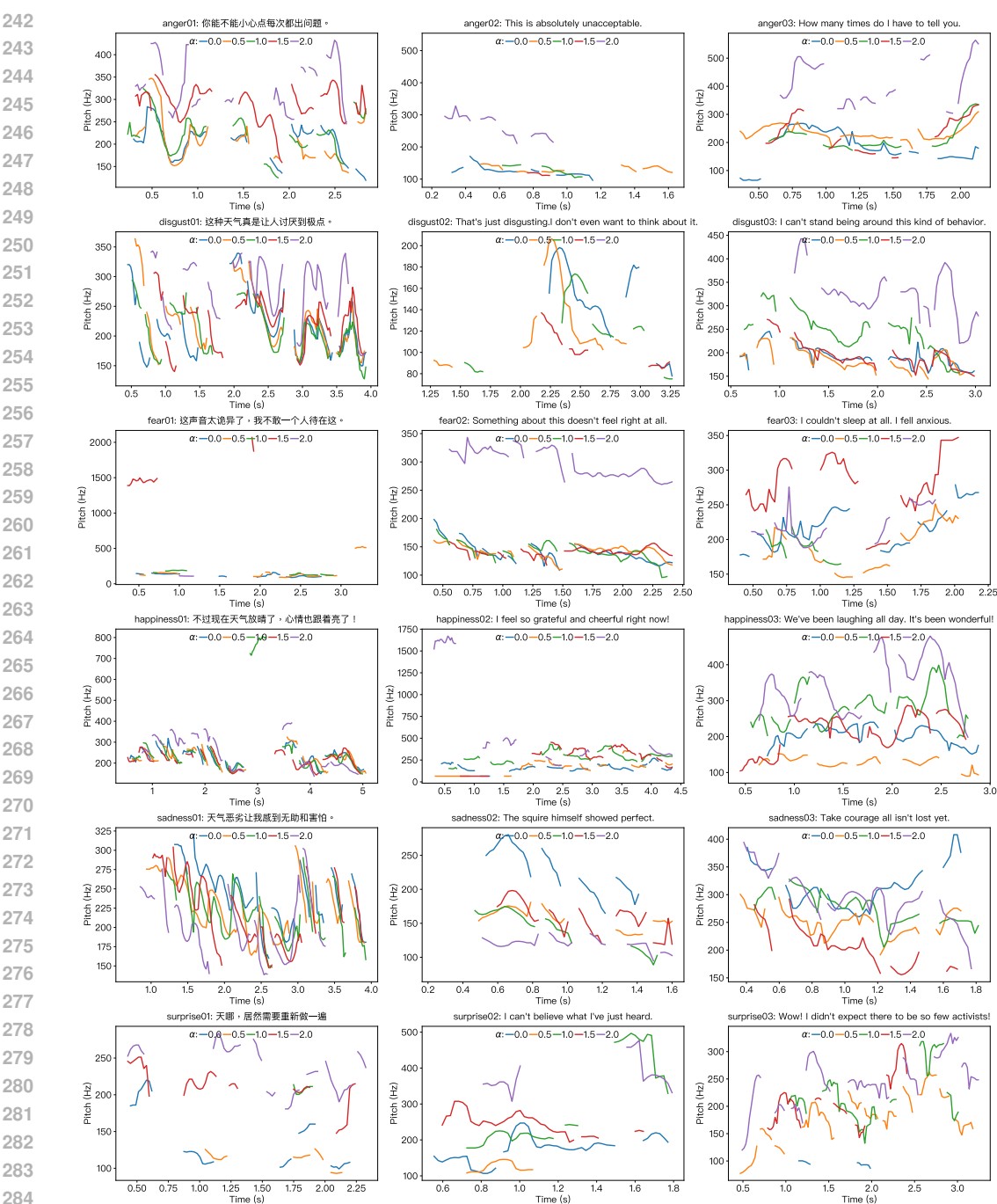

Figure 9: Visualizations of F0 contours in emotion interpolation. From left to right: F5-TTS, E2-TTS, and CosyVoice2. From top to bottom: anger, disgust, fear, happiness, sadness, and surprise. All the synthesized speech samples are interpolated between neutrality ($\alpha$=0) and a target emotion ($\alpha$=2).

## F APPENDIX F: VISUALIZATION OF F0 CONTOURS

### F.1 EMOTION INTERPOLATION

In this subsection, we present additional visualizations of F0 contours to illustrate the fine-grained and continuous emotion interpolation capabilities of the proposed EmoSteer-TTS. As shown in

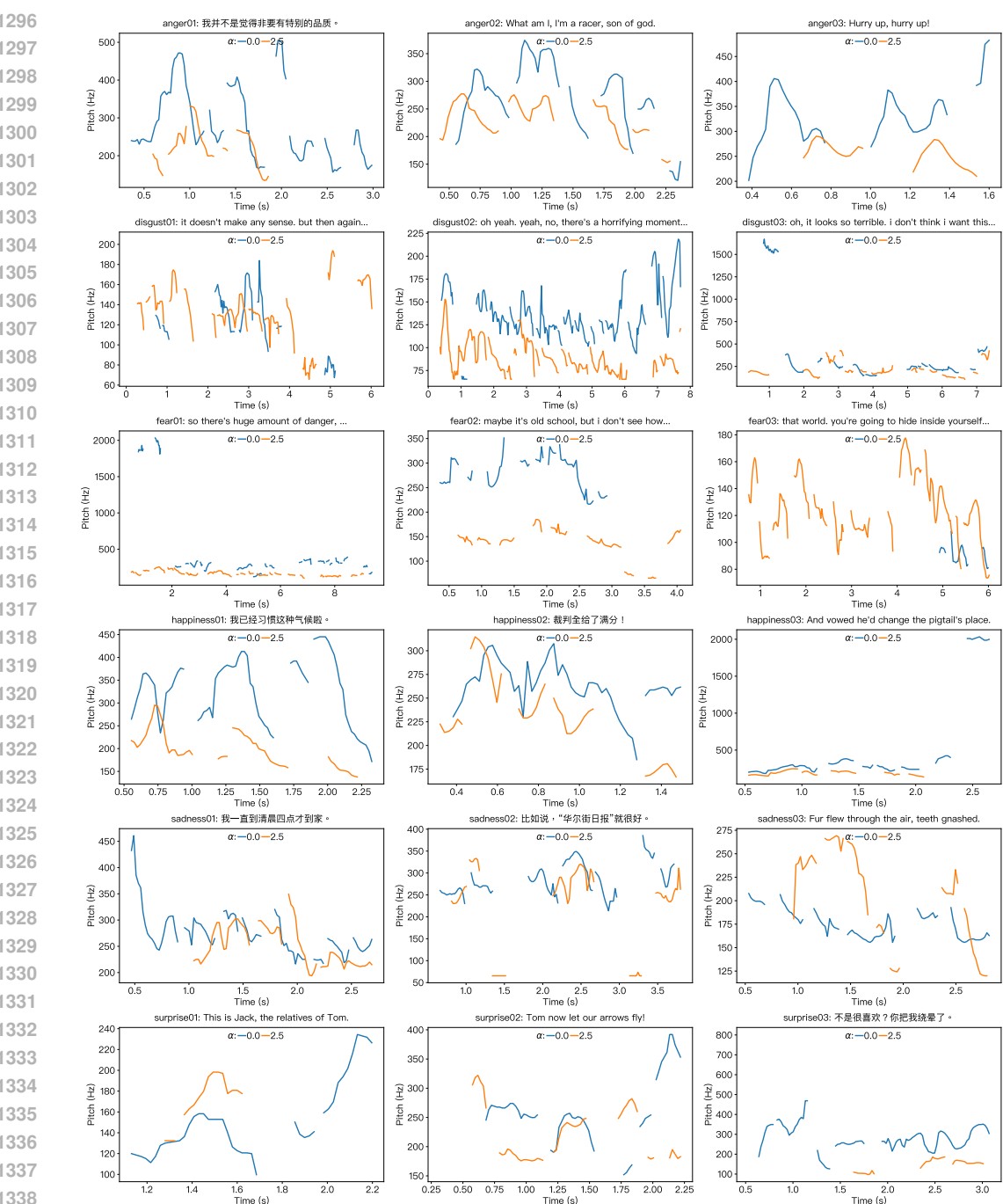

Figure 10: Visualizations of F0 contours in emotion erasure. From left to right: F5-TTS, E2-TTS, and CosyVoice2. From top to bottom: anger, disgust, fear, happiness, sadness, and surprise. All the synthesized speech samples are emotionally erased from a target emotion ($\beta$=0) towards neutrality ($\beta$=2.5).

Fig. 9, voices with angrier, happier, or more surprised tones tend to exhibit higher pitch, while sadder tones are generally associated with lower pitch. In contrast, pitch variations in disgust and fear interpolation show no clear monotonic trend, which we attribute to the fact that these emotions are more closely tied to semantic content than to acoustic characteristics.

Table 6: Emotion conversion ($\alpha = 2.0$) on F5-TTS using emotion2vec for token probing.

| Steering Corpus Composition | WER↓ | S-SIM↑ | E-SIM↑ emotion2vec / SenseVoice | UTMOS↑ |
|---|---|---|---|---|
| 3 datasets (302 samples): IEMOCAP (Busso et al., 2008), SAVEE (Jackson & Haq, 2014), CREMA-D (Cao et al., 2014) | 2.91 | 0.59 | 0.18 / 0.15 | 3.42 |
| 7 datasets (3,021 samples): + MSP-Podcast (Lotfian & Busso, 2017), RAVDESS (Livingstone & Russo, 2018), TESS (Pichora-Fuller & Dupuis, 2020), ASVP-ESD (Landry et al., 2020) | 2.84 | 0.64 | 0.21 / 0.17 | 3.51 |
| 11 datasets (6,900 samples): + CASIA (CASIA, 2023), M3ED (Zhao et al., 2022), ESD (Zhou et al., 2022), Emo-Emilia (Zhao et al., 2025) | 2.79 | 0.64 | 0.29 / 0.26 | 3.49 |

## F.2 Emotion Erasure

In this subsection, we present additional F0 contour visualizations to demonstrate the emotion erasure capability of the proposed EmoSteer-TTS. As shown in Fig. 10, the pitch contours of angry, disgusted, happy, and surprised voices become noticeably flatter after emotion erasure, indicating a calmer prosodic pattern. In contrast, the changes in pitch for fear and sadness are more diverse and less predictable. This variability may stem from the fact that fear can be expressed through multiple vocal styles, such as a low, trembling voice or a high-pitched scream, making it difficult for pitch alone to capture the underlying emotional shift. Similarly, sadness may manifest as either soft weeping or loud crying, resulting in inconsistent pitch patterns that do not reliably reflect emotional intensity.

In both emotion interpolation and erasure, F0 contours capture only a partial aspect of human emotional perception, as pitch alone cannot fully convey complex emotional nuances. Therefore, we encourage readers to listen to the audio samples available on our demo page.

## G Appendix G: The Use of LLMs

Some portions of this paper were paraphrased or refined with the assistance of ChatGPT and Gemini. No content was directly generated by LLMs.

## H Appendix H: Additional Analysis of Emotion Steering Dynamics

### H.1 Sensitivity to Steering Corpus Composition

We conducted an additional ablation study to further examine the sensitivity to the composition of the steering corpus. The entire corpus was constructed from 11 datasets, resulting in a huge number of possible combinations. It is infeasible to evaluate all of them exhaustively. A reasonable strategy is to combine the datasets in chronological order, which may partially reflect overall recording quality as recording devices and speech processing technology improve over time. Therefore, we conduct the ablation using three chronological dataset groups and report WER, S-SIM, E-SIM, and UTMOS on the F5-TTS backbone only.

As shown in Tables 6 and 7, WER, S-SIM, and UTMOS remain largely stable across different steering corpus sizes, indicating that general speech quality and semantic fidelity are minimally affected. In contrast, E-SIM consistently increases with the number of datasets, suggesting that emotion similarity benefits from larger and more diverse steering corpora. Overall, these results indicate that dataset quantity primarily influences emotional control, while other aspects of synthesis are largely insensitive to corpus composition.

Table 7: Emotion erasure ($\beta = 2.5$) on F5-TTS using emotion2vec for token probing.

| Steering Corpus Composition | WER↓ | S-SIM↑ | E-SIM↑ emotion2vec / SenseVoice | UTMOS↑ |
|---|---|---|---|---|
| 3 datasets (302 samples): IEMOCAP (Busso et al., 2008), SAVEE (Jackson & Haq, 2014), CREMA-D (Cao et al., 2014) | 2.88 | 0.61 | 0.07 / 0.05 | 3.51 |
| 7 datasets (3,021 samples): + MSP-Podcast (Lotfian & Busso, 2017), RAVDESS (Livingstone & Russo, 2018), TESS (Pichora-Fuller & Dupuis, 2020), ASVP-ESD (Landry et al., 2020) | 2.94 | 0.58 | 0.18 / 0.12 | 3.68 |
| 11 datasets (6,900 samples): + CASIA (CASIA, 2023), M3ED (Zhao et al., 2022), ESD (Zhou et al., 2022), Emo-Emilia (Zhao et al., 2025) | 2.81 | 0.63 | 0.26 / 0.25 | 3.55 |

Table 8: The Pearson correlation coefficients between the E-SIM (emotion2vec/SenseVoice) scores and the N-MOS and EE-MOS (emotion2vec is used for token probing).

| | E-SIM (emotion2vec) | E-SIM (SenseVoice) |
|---|---|---|
| N-MOS (Conversion, $\alpha = 2.0$) | -0.78 | 0.12 |
| EE-MOS (Erasue, $\beta = 2.5$) | 0.47 | -0.08 |

## H.2 CORRELATION OF E-SIM METRICS WITH N-MOS AND EE-MOS

We report the Pearson correlation coefficients between the E-SIM (emotion2vec/SenseVoice) scores and the N-MOS, EE-MOS ratings for emotion conversion and erasure in our main experiments, respectively. As shown in Table 8, the E-SIM computed with emotion2vec exhibits a clear and consistent trend: it is negatively correlated with N-MOS (–0.78), indicating that stronger steering inevitably leads to noticeable degradation in naturalness. At the same time, it is positively correlated with EE-MOS (+0.47), suggesting that a larger E-SIM (more neutral) corresponds to more successful emotion erasure, as perceived by human raters. This confirms the expected trade-off between emotion controllability and naturalness.

In contrast, the correlations obtained using SenseVoice show almost no relationship with either N-MOS (+0.12) or EE-MOS (–0.08). We attribute this inconsistency to a mismatch between the emotion space captured by SenseVoice and that encoded by emotion2vec, which is also used in our token-probing framework.

## H.3 CONFIDENCE INTERVALS AND SIGNIFICANCE TESTS OF SUBJECTIVE EVALUATION

For the in-distribution evaluation in Table 1, the overall averaged N-MOS, EI-MOS, and EE-MOS across the three backbones, along with their corresponding confidence intervals, are summarized in Table 9. These results indicate that the naturalness of the synthesized speech, the interpolation capability, and the emotion erasure effectiveness of our method are consistently perceived by human raters as "Good" or above.

We conduct significance tests using the N-MOS and EI-MOS ratings from 30 raters, comparing our method with the strongest label-based baselines. We focus on these baselines because they

Table 9: Confidence Intervals of Subjective Evaluation

| Metric | Averaged | Confidence Interval |
|---|---|---|
| N-MOS | 3.42 | 95% of [3.38, 3.46] |
| EI-MOS | 3.65 | 95% of [3.61, 3.69] |
| EE-MOS | 3.86 | 95% of [3.82, 3.90] |

Table 10: Steering strength $\alpha$ vs. N-MOS and WER for F5-TTS.

| $\alpha$ | 0.00 | 0.25 | 0.50 | 0.75 | 1.00 | 1.25 | 1.50 | 2.00 | 2.50 | 3.00 |
|---|---|---|---|---|---|---|---|---|---|---|
| N-MOS (Anger) | 4.27 | 4.24 | 4.25 | 4.18 | 4.02 | 3.93 | 3.64 | 3.41 | 2.60 | 2.15 |
| N-MOS (Disgust) | 4.19 | 4.20 | 4.08 | 4.02 | 3.86 | 3.65 | 3.37 | 3.16 | 2.29 | 2.08 |
| N-MOS (Fear) | 4.32 | 4.22 | 4.13 | 3.92 | 3.68 | 3.42 | 3.37 | 3.28 | 2.36 | 1.93 |
| N-MOS (Happiness) | 4.25 | 4.13 | 4.02 | 3.97 | 3.72 | 3.51 | 3.36 | 3.37 | 1.66 | 1.57 |
| N-MOS (Sadness) | 4.18 | 4.23 | 4.15 | 4.06 | 3.92 | 3.70 | 3.59 | 3.53 | 2.41 | 2.01 |
| N-MOS (Surprise) | 4.22 | 4.16 | 4.08 | 3.91 | 3.78 | 3.49 | 3.35 | 3.37 | 1.80 | 1.59 |
| WER (Anger) | 2.64 | 2.71 | 2.69 | 2.83 | 2.68 | 2.54 | 2.62 | 2.75 | 15.27 | 26.14 |
| WER (Disgust) | 2.81 | 2.47 | 2.92 | 2.58 | 3.11 | 2.73 | 3.05 | 2.66 | 14.83 | 27.42 |
| WER (Fear) | 2.55 | 3.18 | 2.69 | 3.04 | 2.88 | 2.41 | 3.22 | 2.79 | 16.44 | 24.91 |
| WER (Happiness) | 2.93 | 2.62 | 2.85 | 2.50 | 3.07 | 3.29 | 2.74 | 3.18 | 13.97 | 28.33 |
| WER (Sadness) | 2.49 | 2.88 | 3.15 | 2.73 | 2.60 | 3.18 | 2.57 | 3.11 | 15.62 | 25.40 |
| WER (Surprise) | 3.12 | 2.59 | 2.48 | 3.26 | 2.74 | 2.95 | 3.31 | 2.63 | 14.21 | 29.08 |

Table 11: Steering strength $\alpha$ vs. N-MOS and WER for E2-TTS.

| $\alpha$ | 0.00 | 0.25 | 0.50 | 0.75 | 1.00 | 1.25 | 1.50 | 2.00 | 2.50 | 3.00 |
|---|---|---|---|---|---|---|---|---|---|---|
| N-MOS (Anger) | 4.31 | 4.26 | 4.22 | 4.14 | 4.05 | 3.90 | 3.63 | 3.48 | 2.71 | 1.26 |
| N-MOS (Disgust) | 4.27 | 4.22 | 4.05 | 3.99 | 3.82 | 3.69 | 3.33 | 3.12 | 3.04 | 1.94 |
| N-MOS (Fear) | 4.34 | 4.20 | 4.16 | 3.88 | 3.71 | 3.39 | 3.41 | 3.24 | 2.97 | 2.17 |
| N-MOS (Happiness) | 4.23 | 4.15 | 3.98 | 4.00 | 3.63 | 3.56 | 3.31 | 3.35 | 2.62 | 1.85 |
| N-MOS (Sadness) | 4.20 | 4.21 | 4.18 | 4.04 | 3.90 | 3.73 | 3.60 | 3.50 | 2.98 | 2.00 |
| N-MOS (Surprise) | 4.25 | 4.12 | 4.11 | 3.89 | 3.81 | 3.46 | 3.37 | 3.32 | 2.63 | 1.58 |
| WER (Anger) | 3.24 | 3.21 | 3.19 | 3.33 | 3.18 | 3.04 | 3.12 | 3.25 | 15.77 | 25.62 |
| WER (Disgust) | 3.09 | 3.18 | 3.15 | 3.28 | 3.20 | 3.16 | 3.33 | 3.32 | 15.48 | 24.45 |
| WER (Fear) | 3.16 | 3.23 | 3.20 | 3.35 | 3.19 | 3.07 | 3.11 | 3.26 | 11.63 | 25.90 |
| WER (Happiness) | 3.31 | 3.20 | 3.08 | 3.21 | 3.37 | 2.95 | 3.00 | 3.04 | 19.24 | 35.60 |
| WER (Sadness) | 3.15 | 3.22 | 3.19 | 3.34 | 3.21 | 3.08 | 3.14 | 3.28 | 17.70 | 29.00 |
| WER (Surprise) | 3.42 | 3.19 | 3.26 | 3.40 | 3.18 | 3.16 | 3.21 | 3.13 | 16.93 | 29.70 |

provide adjustable emotion intensity control, whereas description-based methods neither support emotion interpolation nor emotion erasure. Specifically, for N-MOS, we compare "EmoSteer-TTS+CosyVoice2" against HED-TTS, and for EI-MOS, we compare "EmoSteer-TTS+F5-TTS" against EmoSphere++.

A two-sided t-test indicates that our method significantly outperforms the baselines, with p-values of $0.01483 < 0.05$ for N-MOS and $0.00732 < 0.01$ for EI-MOS. These results demonstrate that our approach not only preserves naturalness but also more effectively conveys the intended emotion intensity, validating the advantages of our emotion-steering mechanism.

## H.4 TRADE-OFF BETWEEN $\alpha$ AND WER/N-MOS

We have already reported the E-SIM variations in 4(b) for the emotion interpolation experiment. Therefore, using the same synthesized samples and newly synthesized samples with $\alpha = 2.5$, we further present the averaged in-distribution N-MOS and WER variations across the three backbones (F5-TTS, E2-TTS, and CosyVoice2) as a function of the steering strength $\alpha$. For N-MOS, we randomly selected two groups of synthesized samples per emotion per model, where each group contains samples with varying $\alpha$ but identical linguistic content. This design reduces the substantial workload required for human evaluation. The tabulated results are shown in Tables 10, 11, and 12 (WERs are computed using Whisper-Large V3 transcriptions, and the N-MOS scores are averaged across 12 participants).

As shown in Tables 10, 11, and 12, increasing the steering strength $\alpha$ has a very consistent effect across all emotions and all three models. When $\alpha$ is small or moderate (up to about 1.0–1.5), both N-MOS and WER stay close to the baseline, meaning that the emotion direction can be applied without harming speech quality or intelligibility. When $\alpha$ becomes larger, N-MOS gradually drops

Table 12: Steering strength $\alpha$ vs. N-MOS and WER for E2-TTS.

| $\alpha$ | 0.00 | 0.25 | 0.50 | 0.75 | 1.00 | 1.25 | 1.50 | 2.00 | 2.50 | 3.00 |
|---|---|---|---|---|---|---|---|---|---|---|
| N-MOS (Anger) | 4.40 | 4.35 | 4.33 | 4.26 | 4.18 | 4.07 | 3.84 | 3.80 | 2.96 | 1.43 |
| N-MOS (Disgust) | 4.37 | 4.28 | 4.20 | 4.13 | 4.03 | 3.85 | 3.62 | 3.48 | 3.25 | 1.92 |
| N-MOS (Fear) | 4.43 | 4.34 | 4.26 | 4.09 | 3.90 | 3.70 | 3.55 | 3.50 | 3.08 | 2.01 |
| N-MOS (Happiness) | 4.35 | 4.22 | 4.13 | 4.08 | 3.93 | 3.72 | 3.55 | 3.58 | 2.78 | 1.07 |
| N-MOS (Sadness) | 4.30 | 4.32 | 4.21 | 4.15 | 4.05 | 3.84 | 3.66 | 3.62 | 3.16 | 0.98 |
| N-MOS (Surprise) | 4.33 | 4.25 | 4.18 | 4.01 | 3.92 | 3.68 | 3.51 | 3.45 | 2.82 | 1.26 |
| WER (Anger) | 2.65 | 2.72 | 2.70 | 2.84 | 2.69 | 2.55 | 2.63 | 2.77 | 27.58 | 15.92 |
| WER (Disgust) | 2.51 | 2.90 | 2.68 | 2.81 | 2.67 | 2.56 | 2.63 | 2.73 | 18.46 | 28.37 |
| WER (Fear) | 2.67 | 3.14 | 2.71 | 2.87 | 2.71 | 2.59 | 2.64 | 2.78 | 26.25 | 21.73 |
| WER (Happiness) | 2.73 | 3.02 | 2.70 | 2.83 | 2.68 | 2.55 | 2.61 | 2.76 | 15.87 | 29.48 |
| WER (Sadness) | 2.46 | 2.73 | 2.71 | 2.85 | 2.72 | 2.58 | 2.66 | 2.79 | 24.36 | 16.48 |
| WER (Surprise) | 2.62 | 3.09 | 2.67 | 2.82 | 2.70 | 2.56 | 2.63 | 2.74 | 28.79 | 20.11 |

Table 13: Emotion conversion ($\alpha = 2.0$) using SenseVoice for token probing.

| Method | WER↓ | S-SIM↑ | E-SIM↑ emotion2vec / SenseVoice | UTMOS↑ |
|---|---|---|---|---|
| EmoSteer-TTS + F5-TTS | 2.94 | 0.62 | 0.27 / 0.29 | 3.45 |
| EmoSteer-TTS + E2-TTS | 3.46 | 0.60 | 0.25 / 0.26 | 3.26 |
| EmoSteer-TTS + CosyVoice2 | 2.77 | 0.58 | 0.26 / 0.28 | 3.57 |

and WER starts to rise, and extremely large values ($\geq$ 2.5) cause the model to leave its normal operating range and produce distorted speech. This pattern is nearly identical for F5-TTS, E2-TTS, and CosyVoice2, indicating that the behavior is general and that excessive steering can distort the feature representation across all models. This phenomenon may be attributed to shared training practices across the models, e.g., gradient clipping, normalization layers, and other regularization techniques. Therefore, we recommend the following guidance for choosing the steering strength $\alpha$:

- Stable region, less emotional: $\alpha \leq 1.0$

- Controlled, minimal degradation, emotional: $1.0 < \alpha \leq 2.0$

- Unstable region, noisy: $\alpha > 2.0$

### H.5 SENSITIVITY TO THE SER MODEL FOR TOKEN PROBING

Different SER models are trained on different datasets, the final objective scores are therefore also influenced by the particular SER model used to guide the construction of steering vectors.

Specifically, we use SenseVoice for token probing, while reporting E-SIM scores under both emotion2vec and SenseVoice to reveal whether EmoSteer-TTS is overfitting to a specific SER embedding space. We use the same neutral samples from MSP-Podcast and ESD in our main experiments to construct steering vectors and report WER, S-SIM, and E-SIM for emotion conversion ($\alpha = 2.0$) and erasure ($\beta = 2.5$). We also report model-based UTMOS (Saeki et al., 2022) scores instead of N-MOS to avoid the substantial workload associated with human evaluation. The results are shown in the following table.

As shown in Tables 13 and 14, our additional analysis reveals an extremely slight tendency of EmoSteer-TTS to align more closely with the SenseVoice emotion embedding space, as evidenced by the marginally higher E-SIM scores under SenseVoice compared to emotion2vec in Tables 13 and 14. This suggests a mild degree of overfitting to the specific SER model used for token probing.

However, our strong human subjective scores (e.g., EI-MOS and EE-MOS in Table 1) align with the objective metrics, giving us high confidence that EmoSteer-TTS is genuinely effective and not just overfitting to a specific metric's embedding space. We leave for future work the extension to more SER embedding models for selecting the top-k tokens.

Table 14: Emotion erasure ($\beta = 2.5$) using SenseVoice for token probing.

| Method | WER↓ | S-SIM↑ | E-SIM↑ | UTMOS↑ |
|--------|------|--------|--------|--------|
| | | | emotion2vec / SenseVoice | |
| EmoSteer-TTS + F5-TTS | 3.01 | 0.51 | 0.24 / 0.27 | 3.39 |
| EmoSteer-TTS + E2-TTS | 3.67 | 0.49 | 0.23 / 0.23 | 3.18 |
| EmoSteer-TTS + CosyVoice2 | 2.98 | 0.53 | 0.26 / 0.29 | 3.46 |

Table 15: The robustness to noise and reverberation of emotion conversion ($\alpha = 2.0$) on F5-TTS using emotion2vec for token probing.

| | WER↓ | S-SIM↑ | E-SIM↑ | UTMOS↑ |
|--|------|--------|--------|--------|
| | | | emotion2vec / SenseVoice | |
| Noise | 34.27 | 0.47 | 0.18 / 0.16 | 2.94 |
| Reverberation | 12.58 | 0.53 | 0.22 / 0.20 | 3.15 |

## H.6 ROBUSTNESS TO NOISE AND REVERBERATION

To investigate the robustness of our method to speech prompts with noise and reverberation, we collect 100 English samples with noise (from the Microsoft DNS Challenge dataset (Dubey et al., 2024)) and 100 English samples with reverberation (from the REVERB Challenge dataset (Kinoshita et al., 2013)) to report WER, S-SIM, E-SIM, and UTMOS for emotion conversion using F5-TTS backbone.

As shown in Table 15, reverberation has a much smaller impact on emotion conversion than additive noise. Noisy inputs significantly degrade intelligibility (WER = 34.27) and reduce both style and emotional similarity (S-SIM = 0.47, E-SIM = 0.18/0.16). In contrast, reverberant inputs maintain substantially better performance across all metrics (WER = 12.58, S-SIM = 0.53, E-SIM = 0.22/0.20), and also achieve higher perceptual quality (UTMOS = 3.15 vs. 2.94).

Overall, these results indicate that the precomputed steering vector remains robust under moderate reverberation, while strong additive noise introduces more noticeable degradation, although the emotional cues are still partially preserved.

