# OpenReview forum: "EmoSteer-TTS: Fine-Grained and Training-Free Emotion-Controllable Text-to-Speech via Activation Steering"
_ICLR.cc/2026/Conference — ICLR 2026 Conference Withdrawn Submission_

### Official Review · Reviewer_fcfH · 2025-10-28

**Soundness:** 3
**Presentation:** 3
**Contribution:** 3
**Rating:** 6
**Confidence:** 4

**Summary:**

This paper proposes EmoSteer-TTS, a novel framework for fine-grained, continuous emotion control in Text-to-Speech (TTS) models. The key innovation is that this method is training-free; it can be applied to existing, pretrained flow matching-based TTS models (e.g., F5-TTS, CosyVoice2) without any fine-tuning.

The method works by "activation steering." First, the authors compute "steering vectors" by finding the average difference in internal model activations between neutral and emotional speech, using a curated dataset. This vector is then refined by identifying the "top-k" most emotionally salient activation tokens, as validated by an external Speech Emotion Recognition (SER) model. At inference time, this steering vector is added to the model's activations with an adjustable strength parameter (α), allowing for precise control.

**Strengths:**

The core contribution, a training-free, "plug-in" framework for fine-grained emotion control, is novel for EC-TTS. It obviates the need for costly retraining or large, multi-emotion datasets for each new TTS model.

The paper validates EmoSteer-TTS on three different SOTA flow-matching models, demonstrating its general applicability within this model class. The evaluation is robust, combining objective and subjective metrics, and includes an OOD test.

**Weaknesses:**

"Training-Free" vs. "Data-Dependent": The "training-free" claim is slightly misleading. While the TTS model is not trained, the method requires a non-trivial offline process: (1) curating a substantial (6,900-sample) high-quality emotional speech dataset, and (2) using a separate, pretrained SER model (emotion2vec) to process activations and find the top-k tokens. The success of the method is therefore highly dependent on the quality of this curated dataset and the accuracy of the chosen SER model. The sensitivity to these components is not explored.

The authors admit in the limitations (Line 511) and in the text (Line 411) that strong steering (α) can introduce artifacts and unintelligible speech. This is a critical and expected limitation. However, the paper is missing a crucial experiment that quantifies this trade-off. An ablation study plotting steering strength (α) against audio quality (N-MOS) and intelligibility (WER) would be necessary to understand the practical usable range of emotion control.

The method is exclusively demonstrated on flow matching-based TTS models using a DiT backbone. It is an open question whether the core assumption—that emotion is represented in a linearly steerable subspace of activations—generalizes to other dominant TTS architectures, such as VITS (VAE/GAN-based) or codec-based autoregressive models (e.g., VALL-E). This limits the generality of the paper's claims.

**Questions:**

The top-k token selection (Sec 3.3) relies on emotion2vec. How sensitive is the quality of the resulting steering vector to this choice? For instance, what happens if SenseVoice (which was used in the evaluation) is used to generate the steering vectors instead? Does performance degrade?

The claim of composite emotion via linear addition (Eq. 11) is very interesting. Could the authors comment on the qualitative results? Does "anger + sadness" sound like a convincing blend, or does it sound muddled/confused?

---

> ### Author Response · Authors · 2025-11-21
> **Author Response (1/4)**
>
> Thank you for your positive and constructive review. We are pleased you found our core contribution "novel for EC-TTS" and our evaluation "robust." We address your insightful questions and suggestions below.
>
> ---
>
> **[Q1] "Training-Free" vs. "Data-Dependent."**
>
> We appreciate your concern. We agree that training-free does not mean data-free and high-quality data is indeed critical for our method. Therefore, following your suggestion, we conduct additional ablation studies to investigate the influence of the steering corpus. The entire corpus was constructed from 11 datasets, resulting in a huge number of possible combinations. **It is infeasible to evaluate all of them exhaustively. A reasonable strategy is to combine the datasets in chronological order, which may partially reflect overall recording quality as recording devices and speech processing technology improve over time.** Therefore, we conduct the ablation using three chronological dataset groups and report WER, S-SIM, E-SIM, and UTMOS on the F5-TTS backbone only. All sample lists are provided in the supplementary materials. The evaluation results are as follows:
>
> | Steering Corpus Composition                                  | WER$\downarrow$ | S-SIM$\uparrow$ | E-SIM$\uparrow$ (emotion2vec / SenseVoice) | UTMOS$\uparrow$ |
> | ------------------------------------------------------------ | --------------- | --------------- | ------------------------------------------ | --------------- |
> | 3 datasets (302 samples): IEMOCAP (Busso et al., 2008), SAVEE (Jackson & Haq, 2014), CREMA-D (Cao et al., 2014) | 2.91            | 0.59            | 0.18 / 0.15                                | 3.42            |
> | 7 datasets (3,021 samples): + MSP-Podcast (Lotfian & Busso, 2017), RAVDESS (Livingstone & Russo, 2018),  TESS (Pichora-Fuller & Dupuis, 2020), ASVP-ESD (Landry et al., 2020) | 2.84            | 0.64            | 0.21 / 0.17                                | 3.51            |
> | 11 datasets (6,900 samples): + CASIA (CASIA, 2023), M3ED (Zhao et al., 2022), ESD (Zhou et al., 2022), Emo-Emilia (Zhao et al., 2025) | 2.79            | 0.64            | 0.29 / 0.26                                | 3.49            |
>
> Table R6: Emotion conversion ($\alpha=2.0$) on F5-TTS using emotion2vec for token probing.
>
> | Steering Corpus Composition                                  | WER$\downarrow$ | S-SIM$\uparrow$ | E-SIM$\uparrow$ (emotion2vec / SenseVoice) | UTMOS$\uparrow$ |
> | ------------------------------------------------------------ | --------------- | --------------- | ------------------------------------------ | --------------- |
> | 3 datasets (302 samples): IEMOCAP (Busso et al., 2008), SAVEE (Jackson & Haq, 2014), CREMA-D (Cao et al., 2014) | 2.88            | 0.61            | 0.07 / 0.05                                | 3.51            |
> | 7 datasets (3,021 samples): + MSP-Podcast (Lotfian & Busso, 2017), RAVDESS (Livingstone & Russo, 2018),  TESS (Pichora-Fuller & Dupuis, 2020), ASVP-ESD (Landry et al., 2020) | 2.94            | 0.58            | 0.18 / 0.12                                | 3.68            |
> | 11 datasets (6,900 samples): + CASIA (CASIA, 2023), M3ED (Zhao et al., 2022), ESD (Zhou et al., 2022), Emo-Emilia (Zhao et al., 2025) | 2.81            | 0.63            | 0.27 / 0.25                                | 3.55            |
>
> Table R7: Emotion erasure ($\beta=2.5$) on F5-TTS using emotion2vec for token probing.
>
> As shown in Tables R6 and R7, WER, S-SIM, and UTMOS remain largely stable across different steering corpus sizes, indicating that general speech quality and semantic fidelity are minimally affected. In contrast, E-SIM consistently increases with the number of datasets, suggesting that emotion similarity benefits from larger and more diverse steering corpora. Overall, these results indicate that **dataset quantity primarily influences emotional control**, while other aspects of synthesis are largely insensitive to corpus composition. We have added this analysis to **Appendix H.1** in the newly uploaded paper.

---

> ### Author Response · Authors · 2025-11-21
> **Author Response (2/4)**
>
> **[Q2] Artifacts at high strength (**$\alpha$**) and missing trade-off.**
>
> Thank you and this is an excellent point. We agree that understanding the trade-off between steering strength ($\alpha$) and audio quality is crucial for practical use. We have already reported the E-SIM variations in **Fig. 4b** for the **emotion interpolation** experiment. Therefore, using the same synthesized samples and newly synthesized samples with $\alpha=2.5$, we further present the averaged in-distribution N-MOS and WER variations across the three backbones (F5-TTS, E2-TTS, and CosyVoice2) as a function of the steering strength $\alpha$. Due to limited time, for N-MOS, we randomly selected two groups of synthesized samples per emotion per model, where each group contains samples with varying $\alpha$ but identical linguistic content. This design reduces the substantial workload required for human evaluation. The tabulated results are shown below (WERs are computed using Whisper-Large V3 transcriptions, and the N-MOS scores are averaged across 12 participants):
>
> |$\alpha$|0.00|0.25|0.50|0.75|1.00|1.25|1.50|2.00|2.50|3.00|
> |-----------------|----|----|----|----|----|----|----|----|-----|-----|
> |N-MOS(Anger)|4.27|4.24|4.25|4.18|4.02|3.93|3.64|3.41|2.60|2.15|
> |N-MOS(Disgust)|4.19|4.20|4.08|4.02|3.86|3.65|3.37|3.16|2.29|2.08|
> |N-MOS(Fear)|4.32|4.22|4.13|3.92|3.68|3.42|3.37|3.28|2.36|1.93|
> |N-MOS(Happiness)|4.25|4.13|4.02|3.97|3.72|3.51|3.36|3.37|1.66|1.57|
> |N-MOS(Sadness)|4.18|4.23|4.15|4.06|3.92|3.70|3.59|3.53|2.41|2.01|
> |N-MOS(Surprise)|4.22|4.16|4.08|3.91|3.78|3.49|3.35|3.37|1.80|1.59|
> |WER(Anger)|2.64|2.71|2.69|2.83|2.68|2.54|2.62|2.75|15.27|26.14|
> |WER(Disgust)|2.81|2.47|2.92|2.58|3.11|2.73|3.05|2.66|14.83|27.42|
> |WER(Fear)|2.55|3.18|2.69|3.04|2.88|2.41|3.22|2.79|16.44|24.91|
> |WER(Happiness)|2.93|2.62|2.85|2.50|3.07|3.29|2.74|3.18|13.97|28.33|
> |WER(Sadness)|2.49|2.88|3.15|2.73|2.60|3.18|2.57|3.11|15.62|25.40|
> |WER(Surprise)|3.12|2.59|2.48|3.26|2.74|2.95|3.31|2.63|14.21|29.08|
>
> Table R1: Steering strength $\alpha$ vs. N-MOS and WER for F5-TTS.
>
> |$\alpha$|0.00|0.25|0.50|0.75|1.00|1.25|1.50|2.00|2.50|3.00|
> |-----------------|----|----|----|----|----|----|----|----|-----|-----|
> |N-MOS(Anger)|4.31|4.26|4.22|4.14|4.05|3.90|3.63|3.48|2.71|1.26|
> |N-MOS(Disgust)|4.27|4.22|4.05|3.99|3.82|3.69|3.33|3.12|3.04|1.94|
> |N-MOS(Fear)|4.34|4.20|4.16|3.88|3.71|3.39|3.41|3.24|2.97|2.17|
> |N-MOS(Happiness)|4.23|4.15|3.98|4.00|3.63|3.56|3.31|3.35|2.62|1.85|
> |N-MOS(Sadness)|4.20|4.21|4.18|4.04|3.90|3.73|3.60|3.50|2.98|2.00|
> |N-MOS(Surprise)|4.25|4.12|4.11|3.89|3.81|3.46|3.37|3.32|2.63|1.58|
> |WER(Anger)|3.24|3.21|3.19|3.33|3.18|3.04|3.12|3.25|15.77|25.62|
> |WER(Disgust)|3.09|3.18|3.15|3.28|3.20|3.16|3.33|3.32|15.48|24.45|
> |WER(Fear)|3.16|3.23|3.20|3.35|3.19|3.07|3.11|3.26|11.63|25.90|
> |WER(Happiness)|3.31|3.20|3.08|3.21|3.37|2.95|3.00|3.04|19.24|35.60|
> |WER(Sadness)|3.15|3.22|3.19|3.34|3.21|3.08|3.14|3.28|17.70|29.00|
> |WER(Surprise)|3.42|3.19|3.26|3.40|3.18|3.16|3.21|3.13|16.93|29.70|
>
> Table R2: Steering strength $\alpha$ vs. N-MOS and WER for E2-TTS.
>
> |$\alpha$|0.00|0.25|0.50|0.75|1.00|1.25|1.50|2.00|2.50|3.00|
> |-----------------|----|----|----|----|----|----|----|----|-----|-----|
> |N-MOS(Anger)|4.40|4.35|4.33|4.26|4.18|4.07|3.84|3.80|2.96|1.43|
> |N-MOS(Disgust)|4.37|4.28|4.20|4.13|4.03|3.85|3.62|3.48|3.25|1.92|
> |N-MOS(Fear)|4.43|4.34|4.26|4.09|3.90|3.70|3.55|3.50|3.08|2.01|
> |N-MOS(Happiness)|4.35|4.22|4.13|4.08|3.93|3.72|3.55|3.58|2.78|1.07|
> |N-MOS(Sadness)|4.30|4.32|4.21|4.15|4.05|3.84|3.66|3.62|3.16|0.98|
> |N-MOS(Surprise)|4.33|4.25|4.18|4.01|3.92|3.68|3.51|3.45|2.82|1.26|
> |WER(Anger)|2.65|2.72|2.70|2.84|2.69|2.75|2.63|2.77|15.92|27.58|
> |WER(Disgust)|2.51|2.90|2.68|2.81|2.67|2.56|2.63|2.63|18.46|28.37|
> |WER(Fear)|2.67|3.14|2.71|2.87|2.71|2.69|2.54|2.78|21.73|26.25|
> |WER(Happiness)|2.73|3.02|2.70|2.63|2.68|2.55|2.61|2.76|15.87|29.48|
> |WER(Sadness)|2.46|2.73|2.71|2.65|2.72|2.58|2.66|2.69|16.48|24.36|
> |WER(Surprise)|2.62|3.09|2.67|2.82|2.70|2.56|2.63|2.84|20.11|28.79|
>
> Table R3: Steering strength $\alpha$ vs. N-MOS and WER for CosyVoice2.
>
> As shown in Tables R1–R3, increasing the steering strength $\alpha$ has a very consistent effect across all emotions and all three models. **When $\alpha$ is small or moderate (up to about 1.0–1.5), both N-MOS and WER stay close to the baseline**, meaning that the emotion direction can be applied without harming speech quality or intelligibility. **When $\alpha$ becomes larger, N-MOS gradually drops and WER starts to rise, and extremely large values (≥2.5) cause the model to leave its normal operating range and produce distorted speech.** This pattern is nearly identical for F5-TTS, E2-TTS, and CosyVoice2, indicating that the behavior is general and that excessive steering can distort the feature representation across all models. This phenomenon may be attributed to shared training practices across the models, e.g., gradient clipping, normalization layers, and other regularization techniques.
>
> *to be continued ...*

---

> ### Author Response · Authors · 2025-11-21
> **Author Response (3/4)**
>
> **[Q2 Continued**]
>
> Therefore, we recommend the following guidance for choosing the steering strength $\alpha$:
>
> - **Stable region, less emotional:** $\alpha \le 1.0$
> - **Controlled, minimal degradation, emotional:** $1.0 < \alpha \le 2.0$
> - **Unstable region, noisy:** $\alpha > 2.0$
>
> Following your suggestion, we also added this analysis to **Section 4.6** and  **Appendix H.4** in the newly uploaded paper to provide a clear "safe operating range" for users. In future work, we will explore a more unified framework for steering emotions across different models.
>
> ---
>
> **[Q3] Generality limited to flow-matching models.**
>
> We appreciate your concern and agree that this is a valid point regarding the scope of our claims. While our work focuses on and demonstrates effectiveness with the state-of-the-art class of DiT-based flow-matching TTS models, we acknowledge that whether the assumption of a linearly steerable emotion subspace holds for other architectures (e.g., VITS, VAEs, or AR models) remains an open and exciting question. We will **clarify this scope and highlight it as a direction for future work** in the conclusion.
>
> On the other hand, since activation steering has not been explored in the speech domain prior to our work, we believe it can be extended to enable a broader range of applications, including additional types of control and speech-safety alignment. *To this end, we are developing an installable and extensible Python package that supports multiple TTS models across diverse architectures, further benefiting the research community.*
>
> ---
>
> **[Q4] The sensitivity to SER model for probing (top-k)**
>
> Thank you for your question. Following your suggestion, we switch the probing model and the evaluation model to test the robustness of our method.
>
> Specifically, we use **SenseVoice** for token probing, while reporting **E-SIM scores under both emotion2vec and SenseVoice** to reveal whether EmoSteer-TTS is overfitting to a specific SER embedding space. We use the same neutral samples from MSP-Podcast and ESD in our main experiments to construct steering vectors and report WER, S-SIM, and E-SIM for emotion conversion ($\alpha=2.0$) and erasure ($\beta=2.5$). We also report model-based **UTMOS** [1] scores instead of N-MOS to avoid the substantial workload associated with human evaluation. The results are shown in the following table.
>
> |Method|WER$\downarrow$|S-SIM$\uparrow$|E-SIM$\uparrow$(emotion2vec/SenseVoice)|UTMOS$\uparrow$|
> |-------------------------|---------------|---------------|------------------------------------------|---------------|
> |EmoSteer-TTS+F5-TTS|2.94|0.62|0.27/0.29|3.45|
> |EmoSteer-TTS+E2-TTS|3.46|0.60|0.25/0.26|3.26|
> |EmoSteer-TTS+CosyVoice2|2.77|0.58|0.26/0.28|3.57|
>
> Table R4: Emotion conversion ($\alpha=2.0$) using SenseVoice for token probing.
>
> |Method|WER$\downarrow$|S-SIM$\uparrow$|E-SIM$\uparrow$(emotion2vec/SenseVoice)|UTMOS$\uparrow$|
> |-------------------------|---------------|---------------|------------------------------------------|---------------|
> |EmoSteer-TTS+F5-TTS|3.01|0.51|0.24/0.27|3.39|
> |EmoSteer-TTS+E2-TTS|3.67|0.49|0.23/0.23|3.18|
> |EmoSteer-TTS+CosyVoice2|2.98|0.53|0.26/0.29|3.46|
>
> Table R5: Emotion erasure ($\beta=2.5$) using SenseVoice for token probing.
>
> As shown in Tables R4 and R5, our additional analysis reveals an **extremely slight tendency of EmoSteer-TTS to align more closely with the SenseVoice emotion embedding space**, as evidenced by the marginally higher E-SIM scores under SenseVoice compared to emotion2vec in Tables R4 and R5. This suggests a mild degree of overfitting to the specific SER model used for token probing.
>
> However, our **strong human subjective scores** (e.g., EI-MOS and EE-MOS in Table 1 in our paper) align with the objective metrics, giving us high confidence that EmoSteer-TTS is genuinely effective and not just overfitting to a specific metric's embedding space. Due to limited time, we leave for future work the extension to more SER embedding models for selecting the top-k tokens. We added this analysis to **Section 4.7** and  **Appendix H.5** in the newly uploaded paper.

---

> ### Author Response · Authors · 2025-11-21
> **Author Response (4/4)**
>
> **[Q5] Qualitative results of composite emotion**
>
> Thank you for your question. We have provided additional samples for composite emotion steering on the demo site (https://emosteer-tts-demo.pages.dev/). They are generated using neutral reference speech but modified by multi-emotion activation steering. Due to the large number of possible combinations, we provide some examples steered using the "anger + sadness", "anger + disgust", and "fear + surprise" vectors. We kindly ask you to listen to these samples.
>
> From our observations, these samples resemble a linear blend, meaning that the emotion with the stronger steering strength tends to dominate. Achieving more nuanced mixtures by activation manipulation, such as “there is a touch of sadness in the joy,” or “anger intertwined with fear,” remains an open direction for future exploration, e.g., designing novel nonlinear activation steering techniques.
>
> ---
>
> We appreciate your valuable feedback, which we believe has helped strengthen the paper. We are grateful for your support and happy to address any follow-up questions.
>
> We have also uploaded a revised version of our paper, with all newly added or modified content highlighted in blue.

---

> > ### Author Response · Authors · 2025-11-27
> > **Looking forward to your feedback!**
> >
> > Dear Reviewer fcfH,
> >
> > Thank you once again for your valuable feedback. We have conducted additional experiments and made revisions to the paper based on your suggestions. As the discussion phase is nearing its conclusion, we would like to know if our responses have addressed your concerns. We look forward to hearing from you.
> >
> > Best, Authors

---

### Official Review · Reviewer_dQjg · 2025-10-30

**Soundness:** 2
**Presentation:** 2
**Contribution:** 2
**Rating:** 2
**Confidence:** 5

**Summary:**

The paper proposes a training-free method for achieving continuous fine-grained emotion control in TTS synthesis. While the implementation details are mostly provided, several aspects of novelty, clarity, and reproducibility remain unclear. I appreciate the potential contribution toward controllable emotional TTS but there are significant concerns regarding the method’s originality, clarity of presentation, and empirical validation.

**Strengths:**

The idea of training-free fine-grained emotional control is interesting for advancing expressive TTS systems. If it can well explained and validated, the proposed approach has the potential to reduce the reliance on large paired emotional datasets, which remains a challenge in emotional TTS.

**Weaknesses:**

This paper can be further improved by addressing the limitations including insufficient literature coverage, unclear method design, limited novelties, weal results and unclear reproducibility details.

The methodology section is not clearly written. I suggest improving it by explaining the underlying motivation and the rationale behind the design choices. Additionally, please clarify what each equation represents and how it contributes to the overall approach.

The related work section would benefit from including key studies on label-based EC-TTS approaches, as their omission currently makes the authors’ claim less convincing.

I am not fully convinced by the design of the proposed approach, and the paper appears to lack sufficient novelty for ICLR.

The synthesized samples do not clearly reflect effective emotion control, particularly for emotions such as disgust, happiness, sadness, and surprise.

**Questions:**

Do the authors plan to release the curated emotional speech dataset along with the data processing procedures for public use? Additionally, please clarify which testing data and how many utterances were used in the experiments. Would it be possible to evaluate the proposed approach on each dataset separately, and how might this affect its performance?

While the implementation details are mostly described, I wonder whether the authors could release the code to confirm the mathematical details and facilitate reproducibility. If possible, could the authors make it available on the demo page? If not, please provide an explanation.

Why do you think that your approach is the first method that achieves training-free and continuous fine-grained emotion control in TTS? There are several zero-shot TTS models that appear to offer similar capabilities. Please review the related works and provide a fair, detailed comparison against these approaches.

Do the authors plan to release the curated emotional speech dataset along with the data processing procedures for public use? Additionally, please clarify which testing data and how many utterances were used in the experiments. Would it be possible to evaluate the proposed approach on each dataset separately, and how might this affect its performance?
The motivation could be further strengthened by referencing and discussing more recent works on emotional TTS.

What are your novelties for activation steering compared with those in text-to-image models?

The authors claim that label-based approaches rely on paired emotional data, suggesting that their method eliminates this dependency. However, the proposed approach still requires calculating activation differences using paired samples. What distinguishes this method from existing label-based approaches? Moreover, several prior works already achieve comparable results using only a small amount of emotional speech data for training.

---

> ### Author Response · Authors · 2025-11-21
> **Author Response (1/3)**
>
> Thank you for your comments. We believe there are **several fundamental misunderstandings** about our paper's novelty, methodology, evaluation, and reproducibility, which we would like to clarify.
>
> ---
>
> **[Q1] Limited novelty**
>
> Thank you for your comment, which suggests that "zero-shot TTS models... offer similar capabilities" and that our novelty over T2I steering is unclear. This is a key misunderstanding of our contribution. We respectfully but firmly disagree and would like to clarify that:
>
> - **vs. Zero-Shot TTS:** Standard zero-shot TTS models (like our backbones F5-TTS, CosyVoice2) can *clone* an emotion from a reference audio. They cannot perform the core contributions of our work: **fine-grained, continuous emotion interpolation**, **emotion erasure**, or **composite blending**. Compared with prior work, our method **adds** these capabilities to existing uncontrollable models without any retraining. **We would appreciate it if you could point out any relevant references offering similar capabilities we may have overlooked.**
> - **vs. T2I Steering:** While activation steering is a known *concept*, our novelty lies in the **first successful adaptation and application of this technique to the speech synthesis (TTS) domain** for emotion control. The challenges are distinct (e.g., controlling speech emotion vs. static visual concepts). We have also designed a unique emotion-related token searching mechanism for TTS. This is a significant and novel contribution, as recognized by Reviewer C8bf.
> - **Most importantly, our method offers a plug-and-play capability that transforms previously uncontrollable TTS models into emotionally controllable ones without any retraining or fine-tuning,** as recognized by Reviewer fcfH.
> - The novelty is based on these specific, novel capabilities. Therefore, we claim that our method is the first method that achieves training-free and continuous fine-grained emotion control in TTS.
>
> ---
>
> **[Q2] Method clarity & presentation**
>
> Thank you for your comment, which states, "The methodology section is not clearly written" and asks for the rationale behind "unclear math equations". We respectfully but firmly disagree and would like to clarify that:
>
> - We believe our method is presented clearly, as none of the other three reviewers identified clarity as a weakness of our paper. In contrast, **reviewer C8bf, for example, noted our "methodological overview figures" and "concise mathematical formulations" as a strength, making the pipeline "easy to follow."**
> - **Section 3 (Method)** is dedicated to a step-by-step walkthrough of our approach.
> - **Figure 3** provides a clear visual diagram of the entire 3-stage pipeline: (1) Activation Difference Computation, (2) Top-k Emotional Token Searching, and (3) Inference-Time Emotion Steering.
> - To ensure maximum clarity and reproducibility, **Appendix A** provides explicit code snippets for all core operations (conversion, interpolation, erasure, replacement, and multi-steering).
> - **Moreover, your comment in Questions that “the implementation details are mostly described” appears to conflict with your concern regarding the clarity of our method presentation.**
>
> On "please clarify what each equation represents and how it contributes to the overall approach":
>
> - **Sections 3.2, 3.3, and 3.4** provide the mathematical formulations (Eqs. 1-11) and **the rationale for each component** (e.g., L2 norm for stability, SER model for probing, projection for erasure).
>
> - More specifically:
>
>   * Eq. 1 defines Activation Difference.
>   * Eq. 2-4 define Emotional Token Searching.
>   * Eq. 5 defines the Steering Vector.
>   * Eq. 8 & 9 define the Inference-time Control (Conversion/Erasure).
>
> - We are happy to answer specific questions, but "unclear method design" is vague given the mathematical detail provided.
>
> * **We kindly ask you to provide more specific information regarding your clarity concerns, for example, which equation or section you find unclear or hard to understand, and we would be happy to address them.**

---

> ### Author Response · Authors · 2025-11-21
> **Author Response (2/3)**
>
> **[Q3] Training-Free & data-dependent vs. existing label-based methods**
>
> Thank you for your comment, which claims our method is similar to label-based approaches because it uses "paired samples." We believe this is a **misunderstanding** and the distinction is fundamental:
>
> - Label-based methods **train the entire TTS model** on large, paired (text, audio) datasets. Our method indeed needs data to compute steering vectors. **However, we did not suggest or claim that our method is data-free in our paper.**
> - Our method is **"TTS-model-training-free"**. We do not retrain or fine-tune the TTS model at all. We *only* use diverse (neutral, emotional) audio groups instead of text-audio pairs in a **one-time, offline step** to compute a reusable steering vector. This vector then acts as a "plug-in" controller at inference time for *any* text or speaker. This is a fundamental and practical distinction between existing label-based approaches and our method.
>
> On your comment "prior work has achieved comparable results using only a small amount of training data":
>
> - **After re-examining the literature, we did not find any prior label-based work with comparable performance using only a small amount of emotional speech data for training. If convenient, could you kindly provide the relevant references?**
>
> ---
>
> **[Q4] Insufficient evaluation & weak results**
>
> We appreciate your concern. However, we would like to reemphasize that:
>
> - **On evaluation:** We reemphasize that our method's effectiveness is validated by **extensive in-and-out distribution human subjective evaluations** (Table 1), which are the gold standard for TTS and **extensive in-and-out distribution object evaluations (WER, S-SIM, E-SIM)**. Reviewer fcfH also noted that "The evaluation is robust, combining objective and subjective metrics, and includes an OOD test."
> - **On results & synthesized samples:** Our method achieves high scores for Naturalness (N-MOS), Interpolation (EI-MOS), and Erasure (EE-MOS), **outperforming SOTA baselines**. We also score highly on **two independent objective metrics** (emotion2vec and SenseVoice). We strongly encourage the reviewer to listen to the audio samples on our demo page (https://emosteer-tts-demo.pages.dev/), which clearly demonstrate fine-grained control.
> - **If convenient, could you kindly provide more specific details about your concern?** For example, we would appreciate clarification on **which aspect of our evaluation you find insufficient and why**, or **in comparison with which prior work you believe our results appear weak**. We are happy to address your more specific concern.
>
> ---
>
> **[Q5] The coverage of related work is insufficient**
>
> We appreciate your concern. However, **Section 1** and **Section 2** of our paper extensively discuss and cite label-based methods (e.g., EmoSphere++, EmoDubber, HED-TTS) and description-based methods (e.g., EmoVoice, CosyVoice2, PromptTTS). We also directly compare against these SOTA methods in Table 1.
>
> **We kindly ask you to provide more specific details regarding your concern about literature coverage.** **If convenient, could you kindly provide the relevant references?**
>
> ---
>
> **[Q6] Data and code release & reproducibility**
>
> We appreciate your comment, which requests clarification on how the data is used in our experiments and whether we intend to release the dataset and code. You also mentioned that the reproducibility of the paper is unclear. We respectfully but firmly disagree and would like to clarify that:
>
> * **Dataset configuration**: **We kindly and strongly encourage you to read our paper.** Because the dataset composition and split and evaluation protocols are **clearly and comprehensively presented in Section 4 and its subsections for all experiments in the originally submitted paper**, including both an overview and detailed descriptions.
>
> - **Data and code release:** As stated clearly in our **Reproducibility Statement (Section 6, Line 499)** in the first version of our paper, we have already provided the data processing code and dataset in the supplementary material. We apologize, but we are unable to upload the code to our demo site at this time, as it has not yet been cleaned. We will release the fully runnable code and curated dataset upon the paper's acceptance.
> - The code provided in **Appendix A** is the same as that used for activation steering. These steering operations are registered as PyTorch forward hooks, and they are model agnostic.
> - We are also developing an installable and extensible Python package that provides support for additional TTS models, further benefiting the research community.

---

> ### Author Response · Authors · 2025-11-21
> **Author Response (3/3)**
>
> **[Q7] Evaluate the proposed approach on each dataset separately**
>
> **1. Evaluation on different test datasets**
>
> We have provided detailed in-distribution (MSP-Podcast and ESD) and out-of-distribution (SeedTTS and EMNS) evaluations in Sections 4.1–4.5 of our originally uploaded paper.
>
> **2. Evaluation with different steering corpus composition**
>
> To further address your concern, we conduct additional ablation studies to investigate the influence of the steering corpus. The entire corpus was constructed from 11 datasets, resulting in a huge number of possible combinations. **It is infeasible to evaluate all of them exhaustively. A reasonable strategy is to combine the datasets in chronological order, which may partially reflect overall recording quality as recording devices and speech processing technology improve over time.** Therefore, we conduct the ablation using three chronological dataset groups and report WER, S-SIM, E-SIM, and UTMOS on the F5-TTS backbone only. All sample lists are provided in the supplementary materials. The evaluation results are as follows:
>
> | Steering Corpus Composition                                  | WER$\downarrow$ | S-SIM$\uparrow$ | E-SIM$\uparrow$ (emotion2vec / SenseVoice) | UTMOS$\uparrow$ |
> | ------------------------------------------------------------ | --------------- | --------------- | ------------------------------------------ | --------------- |
> | 3 datasets (302 samples): IEMOCAP (Busso et al., 2008), SAVEE (Jackson & Haq, 2014), CREMA-D (Cao et al., 2014) | 2.91            | 0.59            | 0.18 / 0.15                                | 3.42            |
> | 7 datasets (3,021 samples): + MSP-Podcast (Lotfian & Busso, 2017), RAVDESS (Livingstone & Russo, 2018),  TESS (Pichora-Fuller & Dupuis, 2020), ASVP-ESD (Landry et al., 2020) | 2.84            | 0.64            | 0.21 / 0.17                                | 3.51            |
> | 11 datasets (6,900 samples): + CASIA (CASIA, 2023), M3ED (Zhao et al., 2022), ESD (Zhou et al., 2022), Emo-Emilia (Zhao et al., 2025) | 2.79            | 0.64            | 0.29 / 0.26                                | 3.49            |
>
> Table R6: Emotion conversion ($\alpha=2.0$) on F5-TTS using emotion2vec for token probing.
>
> | Steering Corpus Composition                                  | WER$\downarrow$ | S-SIM$\uparrow$ | E-SIM$\uparrow$ (emotion2vec / SenseVoice) | UTMOS$\uparrow$ |
> | ------------------------------------------------------------ | --------------- | --------------- | ------------------------------------------ | --------------- |
> | 3 datasets (302 samples): IEMOCAP (Busso et al., 2008), SAVEE (Jackson & Haq, 2014), CREMA-D (Cao et al., 2014) | 2.88            | 0.61            | 0.07 / 0.05                                | 3.51            |
> | 7 datasets (3,021 samples): + MSP-Podcast (Lotfian & Busso, 2017), RAVDESS (Livingstone & Russo, 2018),  TESS (Pichora-Fuller & Dupuis, 2020), ASVP-ESD (Landry et al., 2020) | 2.94            | 0.58            | 0.18 / 0.12                                | 3.68            |
> | 11 datasets (6,900 samples): + CASIA (CASIA, 2023), M3ED (Zhao et al., 2022), ESD (Zhou et al., 2022), Emo-Emilia (Zhao et al., 2025) | 2.81            | 0.63            | 0.27 / 0.25                                | 3.55            |
>
> Table R7: Emotion erasure ($\beta=2.5$) on F5-TTS using emotion2vec for token probing.
>
> As shown in Tables R6 and R7, WER, S-SIM, and UTMOS remain largely stable across different steering corpus sizes, indicating that general speech quality and semantic fidelity are minimally affected. In contrast, E-SIM consistently increases with the number of datasets, suggesting that emotion similarity benefits from larger and more diverse steering corpora. Overall, these results indicate that **dataset quantity primarily influences emotional control**, while other aspects of synthesis are largely insensitive to corpus composition. We have added this analysis to **Appendix H.1** in the newly uploaded paper.
>
> ---
>
> Thank you again for your comments. We hope these clarifications address your concerns regarding the paper's novelty, methodology, evaluation, and reproducibility, and demonstrate the soundness and significance of our work. We are happy to address any more specific concerns.
>
> We have also uploaded a revised version of our paper, with all newly added or modified content highlighted in blue.

---

### Official Review · Reviewer_C8bf · 2025-11-01

**Soundness:** 3
**Presentation:** 3
**Contribution:** 2
**Rating:** 6
**Confidence:** 2

**Summary:**

This paper proposes EmoSteer-TTS, a training-free method for fine-grained emotion control in text-to-speech via activation steering. The key idea is to identify and manipulate a sparse subset of internal activations in flow matching, DiT-based TTS models (F5-TTS, E2-TTS, CosyVoice2) to modulate emotional tone without additional training. The method constructs per-emotion steering vectors by computing difference-in-means between activations elicited by neutral vs. emotional reference speech, then selecting top-k emotion-relevant token positions using a SER probe to form a weighted steering vector. At inference, the model steers activations with strength α for conversion/interpolation, erases target emotions with strength β via projection-based subtraction, and supports composite operations (replacement or multi-emotion blending). The approach is plug-and-play, adding lightweight hooks to the first residual stream of selected layers and CFM steps.

Empirically, EmoSteer-TTS provides continuous and interpretable control over emotion intensity, outperforming or matching strong baselines on emotion similarity while maintaining intelligibility and speaker similarity, and shows smooth interpolation and effective erasure in both in-distribution and OOD settings. The paper contributes a curated emotion dataset (6,900 utterances) for steering vector construction, detailed implementation hooks, and a thorough analysis of “emotion steering dynamics” across k, layers, and steps. Limitations include reliance on curated emotional references to build steering vectors, potential circularity from using emotion2vec for both probing and evaluation, incomplete fairness in baseline comparisons, limited prosodic analyses beyond F0, artifacts at higher α, and missing efficiency measurements. Overall, the work introduces a novel, practical, and interpretable training-free control mechanism for EC-TTS, with compelling results and clear paths for strengthening evaluation and analysis.

**Strengths:**

* Originality

  - Introduces a training-free, activation-steering paradigm for emotion control in TTS, a clear departure from the prevailing label- or description-conditioned methods that require large-scale training and supervision.
  - Creatively adapts activation steering—previously shown effective in LLMs and T2I diffusion—to flow-matching, DiT-based TTS models, demonstrating cross-domain transfer of a control technique to speech generation.
  - Proposes a principled pipeline to discover emotion-relevant internal tokens: difference-in-means activation extraction between neutral/emotional references, top-k token selection via SER-driven probing, and weighted steering vectors. This yields interpretable, fine-grained control at inference.
  - Expands the control space beyond discrete labels and text prompts to continuous strengths, interpolation between emotional states, erasure of target emotions, and composite manipulation (replacement and multi-emotion blending).
  - Offers analysis of “emotion steering dynamics” across layers, steps, and top-k choices, giving novel insight into how emotion is encoded and can be modulated within flow-matching TTS architectures.

* Quality

  - Extensive empirical evaluation across three strong, diverse backbones (F5-TTS, E2-TTS, CosyVoice2), showing the method is plug-and-play and broadly applicable.
  - Both in-distribution and out-of-distribution tests are reported, with robust performance and minimal degradation, strengthening claims about generalization.
  - Uses multiple complementary metrics: intelligibility (WER), speaker preservation (S-SIM), emotion similarity via two different SER models (emotion2vec and SenseVoice) to reduce metric overfitting risks, and listener studies (N-MOS, EI-MOS, EE-MOS) for perceptual validation.
  - Demonstrates fine-grained control via smooth interpolation curves and F0 contour visualizations; shows effective erasure and composite control with clear quantitative and qualitative evidence.
  - Provides ablations and analyses on top-k, steered layers, and CFM steps, which clarify design choices and the method’s operating regime.
  - Releases detailed implementation hooks and reproducibility details; constructs and describes a curated emotional speech dataset to derive robust steering vectors.

* Clarity

  - The paper is well-structured with clear motivation (limitations of label/prompt-based EC-TTS), methodological overview figures, and concise mathematical formulations of activation extraction, steering, interpolation, erasure, and composite control.
  - Figure 3 and the step-wise algorithm description make the steering pipeline easy to follow; equations use intuitive normalization and projection operations explained in context.
  - Appendices provide code snippets, metric definitions, model configurations, dataset curation, and additional visualizations—substantially improving reproducibility and reader understanding.
  - The limitations are candidly stated (dependency on high-quality emotional samples for steering vector construction, potential artifacts at high steering strengths), which helps situate the claims.

* Significance

  - Addresses a long-standing pain point in EC-TTS—coarse, unstable, and training-heavy control—by enabling stable, continuous, fine-grained emotion manipulation without additional training or large labeled datasets.
  - The training-free, inference-time approach lowers the barrier to adoption: practitioners can retrofit existing TTS models to gain controllability without re-training or data collection, which is impactful for applied scenarios.
  - Composite control (replacement and multi-emotion blending) broadens the expressive repertoire beyond single-label styles, enabling richer user experiences and fine editorial control in applications such as storytelling, assistive agents, and audio post-production.
  - The interpretability of steering vectors and token-level masks provides a new lens to study how emotional tone emerges in TTS models, potentially influencing future architecture designs and control strategies.
  - As the first method (to the authors’ knowledge) achieving training-free, continuous fine-grained emotion control in TTS, EmoSteer-TTS is likely to spur follow-up work at the intersection of controllability, interpretability, and efficiency in speech generation.

**Weaknesses:**

The approach, while training-free at inference, still relies on a curated pool of high-quality emotional speech to build steering vectors, which weakens the claim of being data-free and raises questions about scalability. Please quantify sample complexity (how many and what quality of references are needed), test cross-lingual transfer (build in one language, apply to another), and assess robustness to noise, reverberation, and device/domain mismatch.

Token selection and several evaluations depend on emotion2vec, creating potential circularity and model bias. Beyond adding SenseVoice for evaluation, diversify both probing and evaluation: use multiple heterogeneous SERs (including VAD regressors), include human emotion identification/AB tests with confusion matrices, and explore alternative token-attribution methods (e.g., linear probes, integrated gradients, activation patching, RSA) to ensure the discovered tokens are not artifacts of a single SER.

Comparisons against baselines are weakened by reliance on demo samples and “unguaranteed” reproductions. Re-run competitive training-based and prompt-based EC-TTS under controlled protocols (same text/reference, recommended hyperparameters), and report perceptual results with full details: rater counts, inter-rater reliability, confidence intervals, and significance tests. This is important to substantiate the “first training-free fine-grained control” claim.

The interpretability story—that a sparse subset of tokens encodes emotion—remains preliminary. Provide stronger causal evidence by patching/swapping activations across time/layers, disentangling from phonetic content and speaker traits, and mapping top-k indices to prosodic correlates and VAD dimensions. Time-localization analyses showing whether selected tokens align with prosodic modulation regions would make the findings more convincing.

At higher steering strengths, artifacts appear, and the control lacks principled safeguards. Consider adaptive strength schedules across layers/steps, subspace-constrained steering, or calibration that sets α based on projection magnitude onto the steering vector. Report intelligibility/naturalness as continuous functions of α per backbone to give users safe operating ranges.

Prosodic analysis focuses mainly on F0; this underrepresents emotional variation. Include energy contours, duration/speaking rate, pause statistics, spectral tilt, jitter/shimmer, or eGeMAPS features, and show that erasure drives these toward neutral baselines while interpolation varies them smoothly with α.

Finally, efficiency and composite control need deeper treatment. Quantify runtime/latency overhead (RTF, memory) for different k, layer/step settings, and streaming scenarios, and provide ablations on quality–latency trade-offs. For composite emotions, run targeted listener studies with categorical and dimensional (VAD) ratings to verify that mixtures yield recognizable blends rather than cue superposition; test cross-lingual composite control and references with multiple latent emotions.

**Questions:**

* Data requirements and generalization

How dependent is the method on curated emotional references, and how well do steering vectors transfer across languages, speakers, and acoustic conditions? Clarifying sample needs and robustness to domain shifts would strengthen claims of being training-free and broadly applicable.

* Evaluation methodology and fairness

To what extent do current probes and baselines provide unbiased evidence of effectiveness? A clearer, diversified evaluation (multiple SERs, human studies with rigorous protocols, controlled baseline comparisons) would reduce concerns about circularity and ensure fair positioning against prior work.

* Interpretability and causal validity

The core claim is that a sparse subset of internal tokens encodes emotion and can be causally steered. Can you provide stronger causal tests and time-localization analyses to verify that these tokens specifically modulate emotion (not content or speaker), and relate them to prosodic and VAD dimensions?

* Control stability and safeguards

How do you prevent artifacts and preserve intelligibility/speaker identity at higher steering strengths
alpha (and erasure
beta)? Principles for calibration, adaptive schedules, and clearer safe operating ranges—alongside broader prosodic analyses—would make the control more reliable.

* Practicality, efficiency, and compositionality

What are the runtime/memory costs and scalability to streaming/real-time scenarios, and do composite emotions produce predictable, perceptually coherent blends? Demonstrating efficiency trade-offs and validating compositional control would support real-world adoption.

**Details Of Ethics Concerns:**

Emotion steering in speech synthesis raises several ethical concerns: potential demographic bias (age, gender, accent, language) in perceived emotion intensity, intelligibility, and speaker preservation, especially if steering relies on SER models with known biases; privacy and consent issues around using and releasing voice data, as voice likeness is biometric and may implicate data protection laws and licensing; misuse risks spanning impersonation, social engineering, disinformation, and coercive manipulation, which are heightened by low-friction, real-time control and emotion erasure.

---

> ### Author Response · Authors · 2025-11-21
> **Author Response (1/6)**
>
> We are extremely grateful to your thorough, detailed, and insightful summary of our work. We are delighted that you recognized the originality, quality, clarity, and significance of our contributions. We address your concerns below.
>
> ---
>
> **[Q1] Training-free claim and reliance on curated data**
>
> We appreciate your concern. We agree our method is "data-dependent" for the **one-time offline computation** of steering vectors. We want to clarify that the primary practical benefit of our method is avoiding the costly retraining or fine-tuning of the SOTA TTS model itself. Most importantly, **our method offers a plug-and-play capability that transforms previously uncontrollable TTS models into emotionally controllable ones without any retraining, fine-tuning, and redesign.** Researchers can directly use our precomputed steering vectors to add this functionality to pretrained models.
>
> ---
>
> **[Q2] Sample quantity and quality for steering vector computation**
>
> Thank you for your great question. Following your suggestion, we conducted an additional ablation study to further examine the sensitivity to the composition of the steering corpus. The entire corpus was constructed from 11 datasets, resulting in a huge number of possible combinations. **It is infeasible to evaluate all of them exhaustively. A reasonable strategy is to combine the datasets in chronological order, which may partially reflect overall recording quality as recording devices and speech processing technology improve over time.** Therefore, we conduct the ablation using three chronological dataset groups and report WER, S-SIM, E-SIM, and UTMOS on the F5-TTS backbone only. All sample lists are provided in the supplementary materials. The evaluation results are as follows:
>
> | Steering Corpus Composition                                  | WER$\downarrow$ | S-SIM$\uparrow$ | E-SIM$\uparrow$ (emotion2vec / SenseVoice) | UTMOS$\uparrow$ |
> | ------------------------------------------------------------ | --------------- | --------------- | ------------------------------------------ | --------------- |
> | 3 datasets (302 samples): IEMOCAP (Busso et al., 2008), SAVEE (Jackson & Haq, 2014), CREMA-D (Cao et al., 2014) | 2.91            | 0.59            | 0.18 / 0.15                                | 3.42            |
> | 7 datasets (3,021 samples): + MSP-Podcast (Lotfian & Busso, 2017), RAVDESS (Livingstone & Russo, 2018),  TESS (Pichora-Fuller & Dupuis, 2020), ASVP-ESD (Landry et al., 2020) | 2.84            | 0.64            | 0.21 / 0.17                                | 3.51            |
> | 11 datasets (6,900 samples): + CASIA (CASIA, 2023), M3ED (Zhao et al., 2022), ESD (Zhou et al., 2022), Emo-Emilia (Zhao et al., 2025) | 2.79            | 0.64            | 0.29 / 0.26                                | 3.49            |
>
> Table R6: Emotion conversion ($\alpha=2.0$) on F5-TTS using emotion2vec for token probing.
>
> | Steering Corpus Composition                                  | WER$\downarrow$ | S-SIM$\uparrow$ | E-SIM$\uparrow$ (emotion2vec / SenseVoice) | UTMOS$\uparrow$ |
> | ------------------------------------------------------------ | --------------- | --------------- | ------------------------------------------ | --------------- |
> | 3 datasets (302 samples): IEMOCAP (Busso et al., 2008), SAVEE (Jackson & Haq, 2014), CREMA-D (Cao et al., 2014) | 2.88            | 0.61            | 0.07 / 0.05                                | 3.51            |
> | 7 datasets (3,021 samples): + MSP-Podcast (Lotfian & Busso, 2017), RAVDESS (Livingstone & Russo, 2018),  TESS (Pichora-Fuller & Dupuis, 2020), ASVP-ESD (Landry et al., 2020) | 2.94            | 0.58            | 0.18 / 0.12                                | 3.68            |
> | 11 datasets (6,900 samples): + CASIA (CASIA, 2023), M3ED (Zhao et al., 2022), ESD (Zhou et al., 2022), Emo-Emilia (Zhao et al., 2025) | 2.81            | 0.63            | 0.27 / 0.25                                | 3.55            |
>
> Table R7: Emotion erasure ($\beta=2.5$) on F5-TTS using emotion2vec for token probing.
>
> As shown in Tables R6 and R7, WER, S-SIM, and UTMOS remain largely stable across different steering corpus sizes, indicating that general speech quality and semantic fidelity are minimally affected. In contrast, E-SIM consistently increases with the number of datasets, suggesting that emotion similarity benefits from larger and more diverse steering corpora. Overall, these results indicate that **dataset quantity primarily influences emotional control**, while other aspects of synthesis are largely insensitive to corpus composition. We have added this analysis to **Appendix H.1** in the newly uploaded paper.

---

> ### Author Response · Authors · 2025-11-21
> **Author Response (2/6)**
>
> **[Q3] Generalization ability &  robustness to noise, reverberation, and device/domain mismatch**
>
> Thank you for your question and we respond as follows:
>
> **1. Cross-speaker transfer**
>
> The steering vectors are computed from a high-quality dataset with diverse speakers and content. As our **OOD experiments (Section 4.5, Table 1)** show, these vectors generalize well to unseen speakers, text, and datasets with minimal performance degradation, thereby mitigating the need for model-specific data collection.
>
> **2. Cross-lingual transfer**
>
> This is an interesting question. To examine whether a steering vector computed from one language can transfer to another, we use the precomputed English and Chinese steering vectors and apply them to the same reference samples in our in-distribution emotion conversion experiment to evaluate their cross-lingual transferability. Due to limited time, this experiment is conducted on the F5-TTS backbone only. We report WER, S-SIM, E-SIM, and UTMOS as follows:
>
> |                               | WER$\downarrow$ | S-SIM$\uparrow$ | E-SIM$\uparrow$ (emotion2vec / SenseVoice) | UTMOS$\uparrow$ |
> | ----------------------------- | --------------- | --------------- | ------------------------------------------ | --------------- |
> | English $\rightarrow$ Chinese | 92.74           | 0.21            | 0.13 / 0.08                                | 2.45            |
> | Chinese $\rightarrow$ English | 85.51           | 0.36            | 0.09 / 0.11                                | 3.07            |
>
> Table R9: Cross-lingual emotion conversion ($\alpha=2.0$) on F5-TTS using emotion2vec for token probing.
>
> As shown in Table R9, **cross-lingual transfer of steering vectors remains highly limited**. When applying the English steering vector to Chinese speech (English$\rightarrow$Chinese), WER increases substantially (92.74), and both semantic similarity (S-SIM = 0.21) and emotional similarity (E-SIM = 0.13 / 0.08) drop markedly, indicating degraded linguistic consistency and weak emotional alignment. A similar pattern is observed in the reverse direction (Chinese$\rightarrow$English), though the degradation is slightly milder, with lower WER (85.51) and moderately higher S-SIM (0.36). Overall, the results suggest that **emotion steering directions learned in one language do not directly generalize to another**, likely because the token-to-phoneme mapping, prosodic distributions, and language-specific emotion expression patterns differ across languages. We have added this analysis to **Section 4.7** in our newly uploaded paper.
>
> **3. Robustness to noise, reverberation, and device/domain mismatch**
>
> This is a valid concern. Low-quality samples are not suitable for effective steering vector computation [2-4], and thus we assume you are asking whether a steering vector precomputed from a high-quality dataset remains robust when the reference samples used at inference contain noise, reverberation, or device/domain mismatches. We believe the influence of recording device is negligeeable due to technology advancements nowadays. Therefore,  to answer this question, we collect 100 English samples with noise (from the **Microsoft DNS Challenge dataset** [5]) and 100 English samples with reverberation (from the **REVERB Challenge dataset** [6]) to report WER, S-SIM, E-SIM, and UTMOS for emotion conversion using F5-TTS backbone.
>
> |               | WER$\downarrow$ | S-SIM$\uparrow$ | E-SIM$\uparrow$ (emotion2vec / SenseVoice) | UTMOS$\uparrow$ |
> | ------------- | --------------- | --------------- | ------------------------------------------ | --------------- |
> | Noise         | 34.27           | 0.47            | 0.18 / 0.16                                | 2.94            |
> | Reverberation | 12.58           | 0.53            | 0.22 / 0.20                                | 3.15            |
>
> Table R10: The robustness to noise and reverberation of emotion conversion ($\alpha=2.0$) on F5-TTS using emotion2vec for token probing.
>
> As shown in Table R10, **reverberation has a much smaller impact on emotion conversion than additive noise**. Noisy inputs significantly degrade intelligibility (WER = 34.27) and reduce both style and emotional similarity (S-SIM = 0.47, E-SIM = 0.18/0.16). In contrast, reverberant inputs maintain substantially better performance across all metrics (WER = 12.58, S-SIM = 0.53, E-SIM = 0.22/0.20), and also achieve higher perceptual quality (UTMOS = 3.15 vs. 2.94).
>
> Overall, these results indicate that **the precomputed steering vector remains robust under moderate reverberation**, while **strong additive noise introduces more noticeable degradation**, although the emotional cues are still partially preserved. We have added this analysis to **Appendix H.6** in our newly uploaded paper.

---

> ### Author Response · Authors · 2025-11-21
> **Author Response (3/6)**
>
> **[Q4] Diversify both token probing and performance evaluation**
>
> Thank you for your question and suggestion. We agree that more diverse and comprehensive evaluations would further benefit our method. However, we would like to offer the following clarifications, as some of the proposed analyses would require substantial additional effort, including redesigning our method. Thank you for your understanding:
>
> **1. Using multiple heterogeneous SERs (including VAD regressors) for token selection and evaluation**
>
> This is indeed an interesting direction. However, implementing and validating multiple heterogeneous SER/VAD models would require substantial engineering and computational effort. Due to limited time and scope, we have to leave this as an extension for future work.
>
> **2. Human emotion identification/AB tests with confusion matrices**
>
> We appreciate the suggestion of using confusion matrices. However, such analyses are not directly applicable to generation tasks, as our evaluations rely on subjective ratings rather than discrete class predictions. Instead, we have already included extensive human evaluations (e.g., N-MOS, EI-MOS, and EE-MOS, which are in part analogous to AB tests) in **Table 1** in our paper to comprehensively measure both perceptual quality and controllability.
>
> **3. Explore alternative token-attribution methods**
>
> We agree that exploring more diverse token-probing methodologies is a promising avenue. While the current work focuses on our proposed SER model-based probing, we see this as an exciting future direction and will consider integrating more attribution techniques in subsequent versions.
>
> ---
>
> **[Q5] Baseline comparisons weakened by using demo samples & fairness concern & controlled comparisons**
>
> Thank you for your question, and we would like to clarify the rationale behind using demo samples for comparisons as follows: We appreciate this point and agree that controlled comparisons are vital. In fact, we **included this exact controlled experiment (same text and speech reference) in Appendix E (Table 3).** We found that our own reproductions of open-source baselines (e.g., EmoSphere++) performed significantly worse than their official demos (e.g., WER of 37.29 vs. 16.25). This confirmed our concern that we could not guarantee the quality of reproduced baselines. Therefore, to provide the **strongest and fairest comparison**, we reported the official (and much stronger) demo sample results in our main paper (Table 1), as noted in Line 314 of the originally submitted version.
>
> ---
>
> **[Q6] Interpretability and causal validity**
>
> We appreciate your concern. We agree that a deeper causal investigation would be a non-trivial but exciting extension. We will **highlight this as a promising direction for future work** in our conclusion.
>
> Meanwhile, in Section 4.6, we have conducted a detailed analysis of the impact of top-k tokens on six emotions. As shown in Fig. 7a, increasing k introduces more tokens into the steering signal, potentially capturing a broader range of emotional nuances. Specifically, larger k generally leads to higher average emotion probabilities across all categories, with anger and happiness peaking around k = 200. While incorporating more emotion-relevant tokens enriches the steering signal, the gains plateau beyond k = 200 for most emotions.
>
> ---
>
> **[Q7] Trade-off between $\alpha$ and intelligibility/speaker identity & per backbone safe operating ranges**
>
> Thank you and this is an excellent point. We agree that understanding the trade-off between steering strength ($\alpha$) and audio quality is crucial for practical use. We have already reported the E-SIM variations in **Fig. 4b** for the **emotion interpolation** experiment. Therefore, using the same synthesized samples and newly synthesized samples with $\alpha=2.5$, we further present the averaged in-distribution N-MOS and WER variations across the three backbones (F5-TTS, E2-TTS, and CosyVoice2) as a function of the steering strength $\alpha$. Due to limited time, for N-MOS, we randomly selected two groups of synthesized samples per emotion per model, where each group contains samples with varying $\alpha$ but identical linguistic content. This design reduces the substantial workload required for human evaluation. The tabulated results are shown below (WERs are computed using Whisper-Large V3 transcriptions, and the N-MOS scores are averaged across 12 participants):
>
> *to be continued ...*

---

> ### Author Response · Authors · 2025-11-21
> **Author Response (4/6)**
>
> **[Q7 Continued]**
>
> |$\alpha$|0.00|0.25|0.50|0.75|1.00|1.25|1.50|2.00|2.50|3.00|
> |-----------------|----|----|----|----|----|----|----|----|-----|-----|
> |N-MOS(Anger)|4.27|4.24|4.25|4.18|4.02|3.93|3.64|3.41|2.60|2.15|
> |N-MOS(Disgust)|4.19|4.20|4.08|4.02|3.86|3.65|3.37|3.16|2.29|2.08|
> |N-MOS(Fear)|4.32|4.22|4.13|3.92|3.68|3.42|3.37|3.28|2.36|1.93|
> |N-MOS(Happiness)|4.25|4.13|4.02|3.97|3.72|3.51|3.36|3.37|1.66|1.57|
> |N-MOS(Sadness)|4.18|4.23|4.15|4.06|3.92|3.70|3.59|3.53|2.41|2.01|
> |N-MOS(Surprise)|4.22|4.16|4.08|3.91|3.78|3.49|3.35|3.37|1.80|1.59|
> |WER(Anger)|2.64|2.71|2.69|2.83|2.68|2.54|2.62|2.75|15.27|26.14|
> |WER(Disgust)|2.81|2.47|2.92|2.58|3.11|2.73|3.05|2.66|14.83|27.42|
> |WER(Fear)|2.55|3.18|2.69|3.04|2.88|2.41|3.22|2.79|16.44|24.91|
> |WER(Happiness)|2.93|2.62|2.85|2.50|3.07|3.29|2.74|3.18|13.97|28.33|
> |WER(Sadness)|2.49|2.88|3.15|2.73|2.60|3.18|2.57|3.11|15.62|25.40|
> |WER(Surprise)|3.12|2.59|2.48|3.26|2.74|2.95|3.31|2.63|14.21|29.08|
>
> Table R1: Steering strength $\alpha$ vs. N-MOS and WER for F5-TTS.
>
> |$\alpha$|0.00|0.25|0.50|0.75|1.00|1.25|1.50|2.00|2.50|3.00|
> |-----------------|----|----|----|----|----|----|----|----|-----|-----|
> |N-MOS(Anger)|4.31|4.26|4.22|4.14|4.05|3.90|3.63|3.48|2.71|1.26|
> |N-MOS(Disgust)|4.27|4.22|4.05|3.99|3.82|3.69|3.33|3.12|3.04|1.94|
> |N-MOS(Fear)|4.34|4.20|4.16|3.88|3.71|3.39|3.41|3.24|2.97|2.17|
> |N-MOS(Happiness)|4.23|4.15|3.98|4.00|3.63|3.56|3.31|3.35|2.62|1.85|
> |N-MOS(Sadness)|4.20|4.21|4.18|4.04|3.90|3.73|3.60|3.50|2.98|2.00|
> |N-MOS(Surprise)|4.25|4.12|4.11|3.89|3.81|3.46|3.37|3.32|2.63|1.58|
> |WER(Anger)|3.24|3.21|3.19|3.33|3.18|3.04|3.12|3.25|15.77|25.62|
> |WER(Disgust)|3.09|3.18|3.15|3.28|3.20|3.16|3.33|3.32|15.48|24.45|
> |WER(Fear)|3.16|3.23|3.20|3.35|3.19|3.07|3.11|3.26|11.63|25.90|
> |WER(Happiness)|3.31|3.20|3.08|3.21|3.37|2.95|3.00|3.04|19.24|35.60|
> |WER(Sadness)|3.15|3.22|3.19|3.34|3.21|3.08|3.14|3.28|17.70|29.00|
> |WER(Surprise)|3.42|3.19|3.26|3.40|3.18|3.16|3.21|3.13|16.93|29.70|
>
> Table R2: Steering strength $\alpha$ vs. N-MOS and WER for E2-TTS.
>
> |$\alpha$|0.00|0.25|0.50|0.75|1.00|1.25|1.50|2.00|2.50|3.00|
> |-----------------|----|----|----|----|----|----|----|----|-----|-----|
> |N-MOS(Anger)|4.40|4.35|4.33|4.26|4.18|4.07|3.84|3.80|2.96|1.43|
> |N-MOS(Disgust)|4.37|4.28|4.20|4.13|4.03|3.85|3.62|3.48|3.25|1.92|
> |N-MOS(Fear)|4.43|4.34|4.26|4.09|3.90|3.70|3.55|3.50|3.08|2.01|
> |N-MOS(Happiness)|4.35|4.22|4.13|4.08|3.93|3.72|3.55|3.58|2.78|1.07|
> |N-MOS(Sadness)|4.30|4.32|4.21|4.15|4.05|3.84|3.66|3.62|3.16|0.98|
> |N-MOS(Surprise)|4.33|4.25|4.18|4.01|3.92|3.68|3.51|3.45|2.82|1.26|
> |WER(Anger)|2.65|2.72|2.70|2.84|2.69|2.75|2.63|2.77|15.92|27.58|
> |WER(Disgust)|2.51|2.90|2.68|2.81|2.67|2.56|2.63|2.63|18.46|28.37|
> |WER(Fear)|2.67|3.14|2.71|2.87|2.71|2.69|2.54|2.78|21.73|26.25|
> |WER(Happiness)|2.73|3.02|2.70|2.63|2.68|2.55|2.61|2.76|15.87|29.48|
> |WER(Sadness)|2.46|2.73|2.71|2.65|2.72|2.58|2.66|2.69|16.48|24.36|
> |WER(Surprise)|2.62|3.09|2.67|2.82|2.70|2.56|2.63|2.84|20.11|28.79|
>
> Table R3: Steering strength $\alpha$ vs. N-MOS and WER for CosyVoice2.
>
> As shown in Tables R1–R3, increasing the steering strength $\alpha$ has a very consistent effect across all emotions and all three models. **When $\alpha$ is small or moderate (up to about 1.0–1.5), both N-MOS and WER stay close to the baseline**, meaning that the emotion direction can be applied without harming speech quality or intelligibility. **When $\alpha$ becomes larger, N-MOS gradually drops and WER starts to rise, and extremely large values (≥2.5) cause the model to leave its normal operating range and produce distorted speech.** This pattern is nearly identical for F5-TTS, E2-TTS, and CosyVoice2, indicating that the behavior is general and that excessive steering can distort the feature representation across all models. This phenomenon may be attributed to shared training practices across the models, e.g., gradient clipping, normalization layers, and other regularization techniques. Therefore, we recommend the following guidance for choosing the steering strength $\alpha$:
>
> - **Stable region, less emotional:** $\alpha \le 1.0$
> - **Controlled, minimal degradation, emotional:** $1.0 < \alpha \le 2.0$
> - **Unstable region, noisy:** $\alpha > 2.0$
>
> Following your suggestion, we also added this analysis to **Section 4.6** and **Appendix H.4** in the newly uploaded paper to provide a clear "safe operating range" for users. In future work, we will explore a more unified framework for steering emotions across different models.

---

> ### Author Response · Authors · 2025-11-21
> **Author Response (5/6)**
>
> **[Q8] Prosodic analysis focuses mainly on F0**
>
> Thank you for pointing this out. We agree that F0 can only partially reflect changes in speech emotion, as shown in Fig. 5 in the main text and Figs. 9 and 10 in Appendix F. However, emotions are inherently difficult to quantify, as speakers exhibit different speaking rates and habits, making it challenging to establish a unified standard. **At present, model-based evaluation remains the most reliable approach, such as predicting E-SIM scores as used in our paper.** In future work, we plan to provide more feasible quantitative analyses of emotional variation as you suggested.
>
> ---
>
> **[Q9] Efficiency**
>
> Thank you for your question. To measure the computational efficiency of our method, we use the same settings as in our main experiments (conversion, interpolation, and erasure). The additional average (per sample) inference-time overhead introduced by our method is as follows for each backbone:
>
> | Backbone   | w/o Steering (s) | Emotoin Conversion (s) | Emotoin Interpolation (s) | Emotoin Erasure (s) |
> | ---------- | ---------------- | ---------------------- | ------------------------- | ------------------- |
> | F5-TTS     | 1.867            | 2.415 (**+0.548**)     | 2.504 (**+0.637**)        | 2.746 (**+0.879**)  |
> | E2-TTS     | 0.942            | 1.258 (**+0.316**)     | 1.244 (**+0.302**)        | 1.451 (**+0.509**)  |
> | CosyVoice2 | 3.598            | 4.143 (**+0.545**)     | 4.261 (**+0.663**)        | 4.464 (**+0.866**)  |
>
> Table R11: The Inference time overhead brought by EmoSteer-TTS.
>
> For each type of activation steering, we employ PyTorch hooks to modify the activations during the forward pass. As shown in Table R11, the computational overhead is almost negligible, demonstrating the high efficiency of our methods.
>
> We also added this analysis to **Section 4.8** in the newly uploaded paper.
>
> ---
>
> **[Q10] Compostie control perception**
>
> Thank you for your question. We have provided additional samples for composite emotion steering on the demo site (https://emosteer-tts-demo.pages.dev/). They are generated using neutral reference speech but modified by multi-emotion activation steering. Due to the large number of possible combinations, we provide some examples steered using the "anger + sadness", "anger + disgust", and "fear + surprise" vectors. We kindly ask you to listen to these samples.
>
> From our observations, these samples resemble a linear blend, meaning that the emotion with the stronger steering strength tends to dominate. Achieving more nuanced mixtures by activation manipulation, such as “there is a touch of sadness in the joy.” or “anger intertwined with fear” remains an open direction for future exploration, e.g., designing novel nonlinear activation steering techniques.
>
> ---
>
> **[Q11] Rater counts & reliability, confidence intervals, significance tests**
>
> We appreciate your question and respond as follows:
>
> **1. Rater counts**
>
> 30 raters participated in the human evaluation for our main experiments. All raters were either master's or PhD students.
>
> **2. Inter-rater reliability**
>
> We adopt **Percent Agreement** [7] as a more appropriate measure of reliability for synthesized samples. The results show a **Top-2 Box Agreement of 88.1%**, meaning that the vast majority of ratings fell within the 4 (Good) or 5 (Excellent) categories. Furthermore, the raters demonstrated high consistency in their qualitative judgment, with negligible divergence on the acceptable range.
>
> *to be continued ...*

---

> ### Author Response · Authors · 2025-11-21
> **Author Response (6/6)**
>
> **[Q11 Continued]**
>
> **3. Confidence intervals**
>
> For the in-distribution evaluation in Table 1, the overall averaged N-MOS, EI-MOS, and EE-MOS across the three backbones, along with their corresponding confidence intervals, are summarized in the table below. These results indicate that the naturalness of the synthesized speech, the interpolation capability, and the emotion erasure effectiveness of our method are consistently perceived by human raters as "Good" or above.
>
> | Metric | Averaged | Confidence Interval |
> | ------ | -------- | ------------------- |
> | N-MOS  | 3.42     | 95% of [3.38, 3.46] |
> | EI-MOS | 3.65     | 95% of [3.61, 3.69] |
> | EE-MOS | 3.86     | 95% of [3.82, 3.90] |
>
> Table R12: Confidence Intervals
>
> **4. Significance tests**
>
> We conduct significance tests using the N-MOS and EI-MOS ratings from 30 raters, comparing our method with the strongest label-based baselines. We focus on these baselines because they provide adjustable emotion intensity control, whereas description-based methods neither support emotion interpolation nor emotion erasure. Specifically, for N-MOS, we compare **EmoSteer-TTS+CosyVoice2** against **HED-TTS**, and for EI-MOS, we compare **EmoSteer-TTS+F5-TTS** against **EmoSphere++**.
>
> A two-sided t-test indicates that our method significantly outperforms the baselines, with p-values of **0.01483** < 0.05 for N-MOS and **0.00732** < 0.01 for EI-MOS. These results demonstrate that our approach not only preserves naturalness but also more effectively conveys the intended emotion intensity, validating the advantages of our emotion-steering mechanism.
>
> We have also incorporated these analyses into **Appendix C.3 and Appendix H.3** in the newly uploaded version of our paper.
>
> ---
>
> Thank you again for your meticulous and constructive review. We hope our clarifications and these new analysis can address your concerns and  strengthen the paper, and we are grateful for your support.
>
> We have also uploaded a revised version of our paper, with all newly added or modified content highlighted in blue.
>
> ---
>
> **References**
>
> [1] Saeki, T., Xin, D., Nakata, W., Koriyama, T., Takamichi, S., Saruwatari, H. (2022) UTMOS: UTokyo-SaruLab System for VoiceMOS Challenge 2022. Proc. Interspeech 2022, 4521-4525, doi: 10.21437/Interspeech.2022-439
>
> [2] Yuxin Xiao, Wan Chaoqun, Yonggang Zhang, Wenxiao Wang, Binbin Lin, Xiaofei He, Xu Shen, and Jieping Ye. Enhancing multiple dimensions of trustworthiness in LLMs via sparse activation control. Advances in Neural Information Processing Systems, 37:15730–15764, 2024.
>
> [3] Nithin Gopalakrishnan Nair, Anoop Cherian, Suhas Lohit, Ye Wang, Toshiaki Koike-Akino, Vishal M Patel, and Tim K Marks. Steered diffusion: A generalized framework for plug-and-play conditional image synthesis. In Proceedings of the IEEE/CVF International Conference on Computer Vision, pp. 20850–20860, 2023.
>
> [4] Pau Rodriguez, Arno Blaas, Michal Klein, Luca Zappella, Nicholas Apostoloff, Marco Cuturi, and Xavier Suau. Controlling language and diffusion models by transporting activations. arXiv preprint arXiv:2410.23054, 2024.
>
> [5] Dubey, Harishchandra, Ashkan Aazami, Vishak Gopal, Babak Naderi, Sebastian Braun, Ross Cutler, Alex Ju et al. "Icassp 2023 deep noise suppression challenge." *IEEE Open Journal of Signal Processing* 5 (2024): 725-737.
>
> [6] Kinoshita, Keisuke, Marc Delcroix, Takuya Yoshioka, Tomohiro Nakatani, Emanuel Habets, Reinhold Haeb-Umbach, Volker Leutnant et al. "The REVERB challenge: A common evaluation framework for dereverberation and recognition of reverberant speech." In *2013 IEEE Workshop on Applications of Signal Processing to Audio and Acoustics*, pp. 1-4. IEEE, 2013.
>
> [7] Gwet, Kilem L. *Handbook of inter-rater reliability: The definitive guide to measuring the extent of agreement among raters*. Advanced Analytics, LLC, 2014.

---

> > ### Author Response · Authors · 2025-11-27
> > **Looking forward to your feedback!**
> >
> > Dear Reviewer C8bf,
> >
> > Thank you once again for your valuable feedback. We have conducted additional experiments and made revisions to the paper based on your suggestions. As the discussion phase is nearing its conclusion, we would like to know if our responses have addressed your concerns. We look forward to hearing from you.
> >
> > Best, Authors

---

> ### Author Response · Authors · 2025-11-27
> **Author Response on Ethics Concerns**
>
> Dear AC and Reviewer C8bf, thank you for pointing out the potential ethical concerns. We address them as follows:
>
> **On Bias and Fairness:** We acknowledge that our steering vectors rely on the representations learned by SER models (emotion2vec) and the demographic distribution of our curated dataset. While we utilized 11 diverse corpora to ensure gender balance, the steering vectors are currently language-specific (i.e., for English and Chinese only). Future work will focus on developing language-agnostic steering vectors to ensure equitable performance across accents and dialects.
>
> **On Privacy and Data:** All data used to construct the steering vectors are derived from publicly available, consented academic datasets. As a training-free method, EmoSteer-TTS does not modify model weights, eliminating the risk of accidental memorization of inference-time user data.
>
> **On Misuse and Mitigation:** We acknowledge that fine-grained emotion control increases the realism of synthesized speech, potentially raising the risk of misuse in deepfakes or social engineering. However, our method's interpretability offers a unique advantage: the steering vectors themselves act as known "signatures" of manipulation. To mitigate risks, we strongly advocate for the use of invisible audio watermarking in downstream applications. Furthermore, the "emotion erasure" capability, while potentially misuseable, also serves as a tool for removing toxic emotional cues from speech data used in training safety-aligned models.
>
> We have also incorporated these statements into **Section 7** in the newly uploaded version of our paper.

---

### Official Review · Reviewer_8RhA · 2025-11-03

**Soundness:** 3
**Presentation:** 3
**Contribution:** 2
**Rating:** 4
**Confidence:** 4

**Summary:**

The paper proposes EmoSteer-TTS, a training-free method to achieve fine-grained, continuous, and interpretable emotion control in pretrained flow-matching TTS models (e.g., F5-TTS, E2-TTS, CosyVoice2). The key idea is activation steering: (1) compute activation differences between neutral and target-emotion references; (2) select top-k emotion-relevant tokens to form a steering vector (and weights); and (3) at inference, add the steering vector with strength alpha to chosen layers/steps to modulate synthesis.

**Strengths:**

- Works across multiple flow-matching backbones; no fine-tuning required.
- Empirical analyses give actionable guidance: k≈200 works well; multi-layer (spaced) steering outperforms shallow-only; steering across all flow steps is strongest.
- Maintains performance on EMNS/SeedTTS despite steering vectors built from other corpora.
- Low WER and high speaker similarity versus strong flow-matching baselines.

**Weaknesses:**

- The paper prefers a large alpha but lacks a clear tradeoff curve (alpha vs. WER/N-MOS/E-SIM) and recommended operating range.
- Emotion scores use emotion2vec/SenseVoice; although both are reported, objective metrics can bias toward specific embeddings.

**Questions:**

- How sensitive are results to the steering corpus composition? Any ablation showing performance when removing one corpus from the construction set?
- If emotion2vec and SenseVoice disagree, which correlates better with MOS? Any human-study correlation numbers to justify metric choices?
- Any comparison with AR-based approach regarding the emotional control? A rough comparison would be great.

---

> ### Author Response · Authors · 2025-11-21
> **Author Response (1/4)**
>
> We sincerely thank you for your valuable feedback and "good" ratings on our paper's soundness and presentation. We are encouraged that you recognize our method's strengths and strong performance. We address your main questions below.
>
> ---
>
> **[Q1] Lack of a clear trade-off curve ($\alpha$ vs. WER/N-MOS)**
>
> Thank you and this is an excellent point. We agree that understanding the trade-off between steering strength ($\alpha$) and audio quality is crucial for practical use. We have already reported the E-SIM variations in **Fig. 4b** for the **emotion interpolation** experiment. Therefore, using the same synthesized samples and newly synthesized samples with $\alpha=2.5$, we further present the averaged in-distribution N-MOS and WER variations across the three backbones (F5-TTS, E2-TTS, and CosyVoice2) as a function of the steering strength $\alpha$. Due to limited time, for N-MOS, we randomly selected two groups of synthesized samples per emotion per model, where each group contains samples with varying $\alpha$ but identical linguistic content. This design reduces the substantial workload required for human evaluation. The tabulated results are shown below (WERs are computed using Whisper-Large V3 transcriptions, and the N-MOS scores are averaged across 12 participants):
>
> |$\alpha$|0.00|0.25|0.50|0.75|1.00|1.25|1.50|2.00|2.50|3.00|
> |-----------------|----|----|----|----|----|----|----|----|-----|-----|
> |N-MOS(Anger)|4.27|4.24|4.25|4.18|4.02|3.93|3.64|3.41|2.60|2.15|
> |N-MOS(Disgust)|4.19|4.20|4.08|4.02|3.86|3.65|3.37|3.16|2.29|2.08|
> |N-MOS(Fear)|4.32|4.22|4.13|3.92|3.68|3.42|3.37|3.28|2.36|1.93|
> |N-MOS(Happiness)|4.25|4.13|4.02|3.97|3.72|3.51|3.36|3.37|1.66|1.57|
> |N-MOS(Sadness)|4.18|4.23|4.15|4.06|3.92|3.70|3.59|3.53|2.41|2.01|
> |N-MOS(Surprise)|4.22|4.16|4.08|3.91|3.78|3.49|3.35|3.37|1.80|1.59|
> |WER(Anger)|2.64|2.71|2.69|2.83|2.68|2.54|2.62|2.75|15.27|26.14|
> |WER(Disgust)|2.81|2.47|2.92|2.58|3.11|2.73|3.05|2.66|14.83|27.42|
> |WER(Fear)|2.55|3.18|2.69|3.04|2.88|2.41|3.22|2.79|16.44|24.91|
> |WER(Happiness)|2.93|2.62|2.85|2.50|3.07|3.29|2.74|3.18|13.97|28.33|
> |WER(Sadness)|2.49|2.88|3.15|2.73|2.60|3.18|2.57|3.11|15.62|25.40|
> |WER(Surprise)|3.12|2.59|2.48|3.26|2.74|2.95|3.31|2.63|14.21|29.08|
>
> Table R1: Steering strength $\alpha$ vs. N-MOS and WER for F5-TTS.
>
> |$\alpha$|0.00|0.25|0.50|0.75|1.00|1.25|1.50|2.00|2.50|3.00|
> |-----------------|----|----|----|----|----|----|----|----|-----|-----|
> |N-MOS(Anger)|4.31|4.26|4.22|4.14|4.05|3.90|3.63|3.48|2.71|1.26|
> |N-MOS(Disgust)|4.27|4.22|4.05|3.99|3.82|3.69|3.33|3.12|3.04|1.94|
> |N-MOS(Fear)|4.34|4.20|4.16|3.88|3.71|3.39|3.41|3.24|2.97|2.17|
> |N-MOS(Happiness)|4.23|4.15|3.98|4.00|3.63|3.56|3.31|3.35|2.62|1.85|
> |N-MOS(Sadness)|4.20|4.21|4.18|4.04|3.90|3.73|3.60|3.50|2.98|2.00|
> |N-MOS(Surprise)|4.25|4.12|4.11|3.89|3.81|3.46|3.37|3.32|2.63|1.58|
> |WER(Anger)|3.24|3.21|3.19|3.33|3.18|3.04|3.12|3.25|15.77|25.62|
> |WER(Disgust)|3.09|3.18|3.15|3.28|3.20|3.16|3.33|3.32|15.48|24.45|
> |WER(Fear)|3.16|3.23|3.20|3.35|3.19|3.07|3.11|3.26|11.63|25.90|
> |WER(Happiness)|3.31|3.20|3.08|3.21|3.37|2.95|3.00|3.04|19.24|35.60|
> |WER(Sadness)|3.15|3.22|3.19|3.34|3.21|3.08|3.14|3.28|17.70|29.00|
> |WER(Surprise)|3.42|3.19|3.26|3.40|3.18|3.16|3.21|3.13|16.93|29.70|
>
> Table R2: Steering strength $\alpha$ vs. N-MOS and WER for E2-TTS.
>
> |$\alpha$|0.00|0.25|0.50|0.75|1.00|1.25|1.50|2.00|2.50|3.00|
> |-----------------|----|----|----|----|----|----|----|----|-----|-----|
> |N-MOS(Anger)|4.40|4.35|4.33|4.26|4.18|4.07|3.84|3.80|2.96|1.43|
> |N-MOS(Disgust)|4.37|4.28|4.20|4.13|4.03|3.85|3.62|3.48|3.25|1.92|
> |N-MOS(Fear)|4.43|4.34|4.26|4.09|3.90|3.70|3.55|3.50|3.08|2.01|
> |N-MOS(Happiness)|4.35|4.22|4.13|4.08|3.93|3.72|3.55|3.58|2.78|1.07|
> |N-MOS(Sadness)|4.30|4.32|4.21|4.15|4.05|3.84|3.66|3.62|3.16|0.98|
> |N-MOS(Surprise)|4.33|4.25|4.18|4.01|3.92|3.68|3.51|3.45|2.82|1.26|
> |WER(Anger)|2.65|2.72|2.70|2.84|2.69|2.75|2.63|2.77|15.92|27.58|
> |WER(Disgust)|2.51|2.90|2.68|2.81|2.67|2.56|2.63|2.63|18.46|28.37|
> |WER(Fear)|2.67|3.14|2.71|2.87|2.71|2.69|2.54|2.78|21.73|26.25|
> |WER(Happiness)|2.73|3.02|2.70|2.63|2.68|2.55|2.61|2.76|15.87|29.48|
> |WER(Sadness)|2.46|2.73|2.71|2.65|2.72|2.58|2.66|2.69|16.48|24.36|
> |WER(Surprise)|2.62|3.09|2.67|2.82|2.70|2.56|2.63|2.84|20.11|28.79|
>
> Table R3: Steering strength $\alpha$ vs. N-MOS and WER for CosyVoice2.
>
> *to be continued ...*

---

> ### Author Response · Authors · 2025-11-21
> **Author Response (2/4)**
>
> **[Q1 Continued]**
>
> As shown in Tables R1–R3, increasing the steering strength $\alpha$ has a very consistent effect across all emotions and all three models. **When $\alpha$ is small or moderate (up to about 1.0–1.5), both N-MOS and WER stay close to the baseline**, meaning that the emotion direction can be applied without harming speech quality or intelligibility. **When $\alpha$ becomes larger, N-MOS gradually drops and WER starts to rise, and extremely large values (≥2.5) cause the model to leave its normal operating range and produce distorted speech.** This pattern is nearly identical for F5-TTS, E2-TTS, and CosyVoice2, indicating that the behavior is general and that excessive steering can distort the feature representation across all models. This phenomenon may be attributed to shared training practices across the models, e.g., gradient clipping, normalization layers, and other regularization techniques. Therefore, we recommend the following guidance for choosing the steering strength $\alpha$:
>
> - **Stable region, less emotional:** $\alpha \le 1.0$
> - **Controlled, minimal degradation, emotional:** $1.0 < \alpha \le 2.0$
> - **Unstable region, noisy:** $\alpha > 2.0$
>
> Following your suggestion, we have also added this analysis to **Section 4.6** and **Appendix H.4** in the newly uploaded paper to provide a clear "safe operating range" for users. In future work, we will explore a more unified framework for steering emotions across different models.
>
> ---
>
> **[Q2] Potential bias from using emotion2vec/SenseVoice for evaluation.**
>
> Thank you, and this is a valid and insightful concern. We agree that because different SER models are trained on different datasets, the final objective scores are also influenced by the particular SER model used to guide the construction of steering vectors.
>
> Specifically, we use **SenseVoice** for token probing, while reporting **E-SIM scores under both emotion2vec and SenseVoice** to reveal whether EmoSteer-TTS is overfitting to a specific SER embedding space. We use the same neutral samples from MSP-Podcast and ESD in our main experiments to construct steering vectors and report WER, S-SIM, and E-SIM for emotion conversion ($\alpha=2.0$) and erasure ($\beta=2.5$). We also report model-based **UTMOS** [1] scores instead of N-MOS to avoid the substantial workload associated with human evaluation. The results are shown in the following table.
>
> |Method|WER$\downarrow$|S-SIM$\uparrow$|E-SIM$\uparrow$(emotion2vec/SenseVoice)|UTMOS$\uparrow$|
> |-------------------------|---------------|---------------|------------------------------------------|---------------|
> |EmoSteer-TTS+F5-TTS|2.94|0.62|0.27/0.29|3.45|
> |EmoSteer-TTS+E2-TTS|3.46|0.60|0.25/0.26|3.26|
> |EmoSteer-TTS+CosyVoice2|2.77|0.58|0.26/0.28|3.57|
>
> Table R4: Emotion conversion ($\alpha=2.0$) using SenseVoice for token probing.
>
> |Method|WER$\downarrow$|S-SIM$\uparrow$|E-SIM$\uparrow$(emotion2vec/SenseVoice)|UTMOS$\uparrow$|
> |-------------------------|---------------|---------------|------------------------------------------|---------------|
> |EmoSteer-TTS+F5-TTS|3.01|0.51|0.24/0.27|3.39|
> |EmoSteer-TTS+E2-TTS|3.67|0.49|0.23/0.23|3.18|
> |EmoSteer-TTS+CosyVoice2|2.98|0.53|0.26/0.29|3.46|
>
> Table R5: Emotion erasure ($\beta=2.5$) using SenseVoice for token probing.
>
> As shown in Tables R4 and R5, our additional analysis reveals an **extremely slight tendency of EmoSteer-TTS to align more closely with the SenseVoice emotion embedding space**, as evidenced by the marginally higher E-SIM scores under SenseVoice compared to emotion2vec in Tables R4 and R5. This suggests a mild degree of overfitting to the specific SER model used for token probing.
>
> However, our **strong human subjective scores** (e.g., EI-MOS and EE-MOS in Table 1 in our paper) align with the objective metrics, giving us high confidence that EmoSteer-TTS is genuinely effective and not just overfitting to a specific metric's embedding space. Due to limited time, we leave for future work the extension to more SER embedding models for selecting the top-k tokens. We have added this analysis to **Section 4.7** and  **Appendix H.5** in the newly uploaded paper.

---

> ### Author Response · Authors · 2025-11-21
> **Author Response (3/4)**
>
> **[Q3] Sensitivity to steering corpus composition**
>
> Thank you for your great question. Following your suggestion, we conducted an additional ablation study to further examine the sensitivity to the composition of the steering corpus. The entire corpus was constructed from 11 datasets, resulting in a huge number of possible combinations. **It is infeasible to evaluate all of them exhaustively. A reasonable strategy is to combine the datasets in chronological order, which may partially reflect overall recording quality as recording devices and speech processing technology improve over time.** Therefore, we conduct the ablation using three chronological dataset groups and report WER, S-SIM, E-SIM, and UTMOS on the F5-TTS backbone only. All sample lists are provided in the supplementary materials. The evaluation results are as follows:
>
> | Steering Corpus Composition                                  | WER$\downarrow$ | S-SIM$\uparrow$ | E-SIM$\uparrow$ (emotion2vec / SenseVoice) | UTMOS$\uparrow$ |
> | ------------------------------------------------------------ | --------------- | --------------- | ------------------------------------------ | --------------- |
> | 3 datasets (302 samples): IEMOCAP (Busso et al., 2008), SAVEE (Jackson & Haq, 2014), CREMA-D (Cao et al., 2014) | 2.91            | 0.59            | 0.18 / 0.15                                | 3.42            |
> | 7 datasets (3,021 samples): + MSP-Podcast (Lotfian & Busso, 2017), RAVDESS (Livingstone & Russo, 2018),  TESS (Pichora-Fuller & Dupuis, 2020), ASVP-ESD (Landry et al., 2020) | 2.84            | 0.64            | 0.21 / 0.17                                | 3.51            |
> | 11 datasets (6,900 samples): + CASIA (CASIA, 2023), M3ED (Zhao et al., 2022), ESD (Zhou et al., 2022), Emo-Emilia (Zhao et al., 2025) | 2.79            | 0.64            | 0.29 / 0.26                                | 3.49            |
>
> Table R6: Emotion conversion ($\alpha=2.0$) on F5-TTS using emotion2vec for token probing.
>
> | Steering Corpus Composition                                  | WER$\downarrow$ | S-SIM$\uparrow$ | E-SIM$\uparrow$ (emotion2vec / SenseVoice) | UTMOS$\uparrow$ |
> | ------------------------------------------------------------ | --------------- | --------------- | ------------------------------------------ | --------------- |
> | 3 datasets (302 samples): IEMOCAP (Busso et al., 2008), SAVEE (Jackson & Haq, 2014), CREMA-D (Cao et al., 2014) | 2.88            | 0.61            | 0.07 / 0.05                                | 3.51            |
> | 7 datasets (3,021 samples): + MSP-Podcast (Lotfian & Busso, 2017), RAVDESS (Livingstone & Russo, 2018),  TESS (Pichora-Fuller & Dupuis, 2020), ASVP-ESD (Landry et al., 2020) | 2.94            | 0.58            | 0.18 / 0.12                                | 3.68            |
> | 11 datasets (6,900 samples): + CASIA (CASIA, 2023), M3ED (Zhao et al., 2022), ESD (Zhou et al., 2022), Emo-Emilia (Zhao et al., 2025) | 2.81            | 0.63            | 0.27 / 0.25                                | 3.55            |
>
> Table R7: Emotion erasure ($\beta=2.5$) on F5-TTS using emotion2vec for token probing.
>
> As shown in Tables R6 and R7, WER, S-SIM, and UTMOS remain largely stable across different steering corpus sizes, indicating that **general speech quality and fidelity are minimally affected.** In contrast, E-SIM consistently increases with the number of datasets, suggesting that emotion similarity benefits from larger and more diverse steering corpora. Overall, these results indicate that **dataset quantity primarily influences emotional control**, while other aspects of synthesis are largely insensitive to corpus composition. We have added this analysis to **Appendix H.1** in the newly uploaded paper.

---

> ### Author Response · Authors · 2025-11-21
> **Author Response (4/4)**
>
> **[Q4] Correlation of metrics with MOS**
>
> Following your suggestion, we report the Pearson correlation coefficients between the E-SIM (emotion2vec/SenseVoice) scores and the N-MOS, EE-MOS ratings for emotion conversion and erasure in our main experiments, respectively. As shown in Table R8, the E-SIM computed with **emotion2vec** exhibits a clear and consistent trend: it is **negatively correlated with N-MOS** (–0.78), indicating that stronger steering inevitably leads to noticeable degradation in naturalness. At the same time, it is **positively correlated with EE-MOS** (+0.47), suggesting that a larger E-SIM (more neutral) corresponds to more successful emotion erasure, as perceived by human raters. This confirms the expected trade-off between emotion controllability and naturalness.
>
> In contrast, the correlations obtained using **SenseVoice** show almost no relationship with either N-MOS (+0.12) or EE-MOS (–0.08). We attribute this inconsistency to a mismatch between the emotion space captured by SenseVoice and that encoded by emotion2vec, which is also used in our token-probing framework. We have added this analysis to **Appendix H.2** in the newly uploaded paper.
>
> |                                  | E-SIM (emotion2vec) | E-SIM (SenseVoice) |
> | -------------------------------- | ------------------- | ------------------ |
> | N-MOS (Conversion, $\alpha=2.0$) | -0.78               | 0.12               |
> | EE-MOS (Erasue, $\beta=2.5$)     | 0.47                | -0.08              |
>
> Table R8: The Pearson correlation coefficients between the E-SIM (emotion2vec/SenseVoice) scores and the N-MOS and EE-MOS (emotion2vec is used for token probing).
>
> ---
>
> **[Q5] Comparison with AR-based approaches**
>
> Thank you for your question. In fact, EmoVoice, CosyVoice2, and FleSpeech all generate speech tokens autoregressively, which are then passed through a flow-matching model to synthesize mel-spectrograms. However, these methods rely on text prompts and lack mechanisms for fine-grained control. We will support a wider range of TTS models in our open-source, installable Python package, which will be released in several months.
>
> ---
>
> Thank you again! We hope these clarifications and the new $\alpha$ trade-off and metric correlation analysis fully address your concerns and demonstrate the significance of our contribution. We are happy to reply your follow-up questions.
>
> We have also uploaded a revised version of our paper, with all newly added or modified content highlighted in blue.
>
> ---
>
> **References**
>
> [1] Saeki, T., Xin, D., Nakata, W., Koriyama, T., Takamichi, S., Saruwatari, H. (2022) UTMOS: UTokyo-SaruLab System for VoiceMOS Challenge 2022. Proc. Interspeech 2022, 4521-4525, doi: 10.21437/Interspeech.2022-439

---

> > ### Author Response · Authors · 2025-11-27
> > **Looking forward to your feedback!**
> >
> > Dear Reviewer 8RhA,
> >
> > Thank you once again for your valuable feedback. We have conducted additional experiments and revised the paper based on your suggestions. As the discussion phase is nearing its conclusion, we would like to know whether our responses have adequately addressed your concerns. We look forward to hearing from you.
> >
> > Best, Authors

---

### Author Response · Authors · 2025-11-23
**Global Response: Summary of Rebuttals, Paper Updates, and Contributions**

Dear ACs and Reviewers,

We sincerely thank you for your thorough and constructive feedback. We appreciate the time and effort you put into helping us improve our work. We are truly grateful for the many **positive comments** we have received:

* *“Introduces a novel, practical, and interpretable training-free control mechanism for EC-TTS, with compelling results”, “Expands the control space beyond discrete labels... to continuous strengths, interpolation... and composite manipulation”, “Well-structured with clear motivation... and concise mathematical formulations”, “Likely to spur follow-up work at the intersection of controllability, interpretability, and efficiency in speech generation.”*  (**C8bf**)
* *“The core contribution, a training-free, 'plug-in' framework for fine-grained emotion control, is novel for EC-TTS”, “Validates EmoSteer-TTS on three different SOTA flow-matching models, demonstrating its general applicability”, “The evaluation is robust, combining objective and subjective metrics, and includes an OOD test.”*  (**fcfH**)
* *“Works across multiple flow-matching backbones; no fine-tuning required”, “Low WER and high speaker similarity versus strong flow-matching baselines”, “Empirical analyses give actionable guidance.”*  (**8RhA**)
* *“The idea of training-free fine-grained emotional control is interesting for advancing expressive TTS systems”, “Has the potential to reduce the reliance on large paired emotional datasets.”*  (**dQjg**)

The reviewers have therefore highlighted the novelty and practicality of our training-free framework, the robust generalization across multiple backbones, and the effectiveness of the fine-grained control capabilities.

---
## Paper Updates

We have conducted extensive additional experiments and analyses to strengthen our paper. The revised paper has been uploaded with all changes highlighted in blue. The core modifications and additional analyses are:

- **Trade-off Analysis and Safe Operating Range:** We conducted a comprehensive analysis of the trade-off between steering strength ($\alpha$) and audio quality (N-MOS)/intelligibility (WER) across all three backbones (F5-TTS, E2-TTS, and CosyVoice2). Based on these results, we provided specific guidance on the "safe operating range" for users (e.g., Stable region $\alpha \le 1.0$, Controlled region $1.0 < \alpha \le 2.0$).

- **Metric Bias and Correlation Analysis:** To address concerns about potential bias in our evaluation metrics, we introduced **SenseVoice** (in addition to emotion2vec) for token probing and evaluation, confirming that our method is not overfitting to a specific embedding space. We also reported the Pearson correlation coefficients between objective metrics (E-SIM) and human subjective ratings (N-MOS/EE-MOS), confirming the expected trade-off between controllability and naturalness.

- **Steering Corpus Sensitivity:** We performed an ablation study on the composition of the steering corpus (using 3, 7, and 11 datasets). The results indicate that while a larger and more diverse corpus improves emotion similarity (E-SIM), the general speech quality and fidelity remain largely stable.

- **Robustness and Generalization:** We evaluated the robustness of precomputed steering vectors against **noise and reverberation**, finding that the method remains robust under moderate reverberation. We also investigated **cross-lingual transferability**, which we found to be limited, suggesting language-specific emotion patterns.

- **Efficiency Analysis:** We quantified the computational efficiency of our method, demonstrating that the inference-time overhead introduced by our method is negligible across all backbones.

- **Composite Emotion Demonstrations:** We have updated our demo site (https://emosteer-tts-demo.pages.dev/) with new audio samples demonstrating **composite emotion steering** (e.g., "anger + sadness").

- **Clarifications on Novelty and Methodology:** We have refined our paper to clarify the distinction between our "training-free" approach and existing methods: **Our paper proposes the first method that offers a plug-and-play capability that transforms pretrained uncontrollable TTS models into emotionally controllable ones without any retraining or fine-tuning.**

---
# ⚠️
As noted by **Area Chair bJJ4**, we have also responded to many **Misunderstandings** and **Factual Errors** from reviewer **dQjg**.

---
We summarize our **contributions** to the speech domain as follows:
* To the best of our knowledge, this is the **first** paper to introduce activation steering into the speech domain.
* Our results provide clear evidence that speech characteristics can be systematically manipulated by modifying the internal activations of a pre-trained speech model.
* Our work points to promising future directions, such as enhancing the safety (e.g., by erasing harmful speech content without training or fine-tuning) of speech language models through activation steering.

Thank you again!

Authors

---

### Note · Authors · 2026-01-29

I have read and agree with the venue's withdrawal policy on behalf of myself and my co-authors.

---

### Meta-Review · Area_Chair_RiYW · 2026-01-09

**Summary:**

This paper was reviewed by four experts in the field. The recommendations are (2, 4, 6, 6). The reviewers agree that the paper's quality is insufficient for publication and needs significant revision and careful polishing (experimental evaluation, paper presentation, comparison, etc.). Nearly all of the four Reviewers hold serious concerns that the experimental evaluations might be biased, and more detailed ablation studies and objective experimental evaluations are needed to make the proposed solution and design more convincing. Besides, Reviewer dQjg also argues some more limitations, including insufficient literature coverage, unclear method design, limited novelties, weak results, and unclear reproducibility details. The manuscript needs to be further polished to make the motivation clearer and more convincing. Taking these concerns into consideration, the paper would not be accepted at this time. The authors are encouraged to consider the reviewers' comments when revising the paper for submission elsewhere.

**Reviewer Concerns:**

The authors have successfully addressed concerns regarding specific comparisons, discussions, and the paper's presentation in their rebuttal. However, the experimental evaluation remains a significant shortcoming. Despite the revisions, the additional comparisons and evaluations provided are insufficient. Consequently, the empirical evidence fails to robustly validate the proposed method's superiority over existing baselines.

**Reviewer Scores:**

The reviewers would be satisfied with the author's commitment to adding more result comparisons and discussions, and polishing the paper presentation in the revised version, while they may still hold several concerns regarding the provided experimental evaluations and in-depth analysis.

---

### Decision · Program_Chairs · 2026-01-26

Reject